# Complementary α-arrestin-ubiquitin ligase complexes control nutrient transporter endocytosis in response to amino acids

Vasyl Ivashov[1†], Johannes Zimmer[1†], Sinead Schwabl[1], Jennifer Kahlhofer[1], Sabine Weys[1], Ronald Gstir[2], Thomas Jakschitz[2], Leopold Kremser[3], Günther K Bonn[2], Herbert Lindner[3], Lukas A Huber[1,2], Sebastien Leon[4], Oliver Schmidt[1*], David Teis[1]

[1]Institute for Cell Biology, Medical University of Innsbruck, Innsbruck, Austria; [2]ADSI – Austrian Drug Screening Institute GmbH, Innsbruck, Austria; [3]Division of Clinical Biochemistry, ProteinMicroAnalysis Facility, Medical University of Innsbruck, Innsbruck, Austria; [4]Université de Paris, CNRS, Institut Jacques Monod, Paris, France

**Abstract** How cells adjust nutrient transport across their membranes is incompletely understood. Previously, we have shown that *S. cerevisiae* broadly re-configures the nutrient transporters at the plasma membrane in response to amino acid availability, through endocytosis of sugar- and amino acid transporters (AATs) (Müller et al., 2015). A genome-wide screen now revealed that the selective endocytosis of four AATs during starvation required the α-arrestin family protein Art2/Ecm21, an adaptor for the ubiquitin ligase Rsp5, and its induction through the general amino acid control pathway. Art2 uses a basic patch to recognize C-terminal acidic sorting motifs in AATs and thereby instructs Rsp5 to ubiquitinate proximal lysine residues. When amino acids are in excess, Rsp5 instead uses TORC1-activated Art1 to detect N-terminal acidic sorting motifs within the same AATs, which initiates exclusive substrate-induced endocytosis. Thus, amino acid excess or starvation activate complementary α-arrestin-Rsp5-complexes to control selective endocytosis and adapt nutrient acquisition.

*For correspondence:
oliver.schmidt@i-med.ac.at

†These authors contributed equally to this work

Competing interests: The authors declare that no competing interests exist.

## Introduction

Cells regulate the import of amino acids, glucose, and other nutrients to fuel metabolism, sustain growth or maintain homeostasis. For the selective transport of amino acids across the plasma membrane (PM) and other cellular membranes, the human genome encodes more than 60 known amino acid transporters (AATs). Mutations in AATs cause severe defects of amino acid metabolism, and the deregulation of AATs is linked to a range of human pathologies, including neurodegenerative diseases, diabetes and cancer (*Kandasamy et al., 2018*; *Smith, 1990*; *McCracken and Edinger, 2013*; *Zhang et al., 2017*).

AATs belong to the family of solute carriers (SLCs) and form selective pores that change from an outward- to an inward-facing conformation and thereby transport their amino acid substrates across membranes. Hence, the addition of AATs to the PM or their selective removal by endocytosis determines quantity and quality of amino acid transport. In human cells, the molecular mechanisms leading to the selective endocytosis of AATs are largely unclear.

The mechanisms for the endocytic down-regulation of AATs are beginning to emerge in the budding yeast, *S. cerevisiae*. Yeast cells frequently experience acute fluctuations in amino acid availability (*Broach, 2012*) and can rapidly remodel their 21 PM-localized AATs to optimize the import of amino acids with regard to the quantity and quality of the nitrogen source available (*Grenson et al.,*

1966; *Grenson, 1966*; *André, 2018*; *Van Belle and André, 2001*; *Bianchi et al., 2019*). The selective, ubiquitin-dependent endocytosis of AATs and other integral PM proteins frequently requires the HECT-type ubiquitin ligase Rsp5, the orthologue of human Nedd4 (*Hein et al., 1995*; *Dupré et al., 2004*; *Belgareh-Touzé et al., 2008*). To specifically recognize and ubiquitinate PM proteins, Rsp5 interacts with the α-arrestin proteins, a family of ubiquitin ligase adaptors (*Nikko et al., 2008*; *Lin et al., 2008*). The budding yeast genome encodes 14 α-arrestin proteins (arrestin-related trafficking adaptors, ARTs; Art1-10, Bul1-3 and Spo23), which function in a partially redundant manner (*Becuwe et al., 2012a*; *MacGurn et al., 2012*; *Babst, 2020*; *O'Donnell and Schmidt, 2019*; *Nikko and Pelham, 2009*). Six α-arrestins have been identified in humans (*Alvarez, 2008*). α-Arrestin proteins use PPxY (PY) motifs to interact directly with the three WW domains of Rsp5/Nedd4 (*Becuwe et al., 2012a*; *Rauch and Martin-Serrano, 2011*). How α-arrestins recognize their cargoes in general, and more specifically AATs, is only partially understood, but appears to involve their arrestin domains. These domains have large, disordered loop- and tail-like insertions that contribute to the function of ARTs (*Baile et al., 2019*).

The activity of α−arrestins is controlled by complex post-translational modifications (PTMs) involving (de-)ubiquitination and (de-)phosphorylation. In several cases, the binding of α−arrestins to Rsp5 results in their ubiquitination, which is required for function (*Lin et al., 2008*; *Becuwe et al., 2012b*; *Hovsepian et al., 2017*). In case of Art1, ubiquitination determines its localization (*Lin et al., 2008*). Additional ubiquitination of ARTs can target them for proteasomal degradation, which is counteracted by deubiquitinating enzymes (*Ho et al., 2017*). Several α−arrestins are activated by dephosphorylation events in response to nutrient availability or other cellular stresses (*Becuwe et al., 2012a*; *O'Donnell et al., 2013*; *Alvaro et al., 2014*; *Becuwe et al., 2012b*; *Becuwe and Léon, 2014*; *Merhi and André, 2012*). One example are high levels of amino acids, which activate the target of rapamycin complex 1 (TORC1). This leads to inhibition of the kinase Npr1 and thereby promotes the activity of Art1 (*MacGurn et al., 2011*). At the PM, Ppz phosphatases activate Art1 by dephosphorylation of its Npr1-dependent phosphorylation sites (*Lee et al., 2019*) and thereby stimulate ubiquitination and endocytosis of the Art1 cargoes Mup1 (methionine transporter), Can1 (arginine transporter), Tat2 (tyrosine and tryptophan transporter) and Lyp1 (lysine transporter) in response to substrate excess. Similarly, TORC1 signaling also promotes endocytosis of the general amino acid permease (Gap1) via the ARTs Bul1/2 (*Merhi and André, 2012*). It seems that nutrient-dependent TORC1 activation controls α−arrestin-mediated ubiquitin-dependent AAT endocytosis to adjust amino acid influx to metabolic needs.

Activated Art1 recognizes specific sorting signals in AATs. The flux of methionine through Mup1 or arginine through Can1 requires conformational changes into the inward-facing conformation, which probably dislodge a C-terminal plug that otherwise 'seals' the pore. At the same time, it exposes an N-terminal acidic patch that is recognized by activated Art1. Art1 interacts with this acidic patch and then orients Rsp5 to ubiquitinate nearby lysine residues (*Guiney et al., 2016*; *Gournas et al., 2018*; *Busto et al., 2018*; *Ghaddar et al., 2014*). Similar results have been obtained for the uracil transporter Fur4 (*Keener and Babst, 2013*). A further layer of complexity may be added by stimulus-induced phosphorylation of nutrient transporters (*Marchal et al., 1998*; *Paiva et al., 2009*), akin to the recognition of G-protein-coupled receptors (GPCRs) by β−arrestins in mammalian cells (*Nikko et al., 2008*; *Nobles et al., 2011*). Hence, a molecular basis of α−arrestin-dependent nutrient transporter endocytosis in response to excess nutrients is emerging. We refer to this process as substrate-induced endocytosis.

Yet, not only substrate transport can induce nutrient transporter endocytosis, but also nutrient limitation, acute starvation or rapamycin treatment (*Crapeau et al., 2014*; *Hovsepian et al., 2017*; *Jones et al., 2012*; *Laidlaw et al., 2020*; *Lang et al., 2014*; *Müller et al., 2015*; *Schmidt et al., 1998*; *Yang et al., 2020*). Starvation-induced endocytosis is part of a coordinated catabolic cascade. Together with proteasomal degradation and with macro- and micro-autophagy, it maintains amino acid homeostasis to allow entry into quiescence for cell survival during nutrient limitation. This catabolic cascade at the onset of starvation appears to be conserved in metazoans (*Mejlvang et al., 2018*; *Edinger and Thompson, 2002a*; *Edinger and Thompson, 2002b*; *Vabulas and Hartl, 2005*; *Suraweera et al., 2012*).

The molecular mechanisms of starvation-induced endocytosis are not clear. Hence, there is a significant knowledge gap in understanding how cells control nutrient transporter abundance upon amino acid and nitrogen scarcity. Here, we have characterized how amino acid abundance alters

several nutrient transporters at the PM of the budding yeast *S. cerevisiae*. We find that TORC1 signaling and the general amino acid control (GAAC) pathway toggle Art1- or Art2-Rsp5 complex activities to induce endocytosis of the same set of four different AATs depending on amino acid availability. In these AATs, activated Art1- or Art2-Rsp5 complexes recognize distinct acidic sorting signals in the N-terminal (Art1 sorting signal) or C-terminal (Art2 sorting signal) regions to initiate Rsp5-dependent ubiquitination and subsequent endocytic down-regulation. Using such complementary α−arrestin switches in combination with distinct acidic sorting motifs in a single nutrient transporter could represent a more general mechanism to adjust transport across cellular membranes and thereby to meet metabolic demands.

## Results

### Amino acid availability induces selective endocytosis of nutrient transporters

We employed *S. cerevisiae* as a model system to address how eukaryotic cells adjust their nutrient transporters at the plasma membrane (PM) to nutrient availability. First, we used live cell fluorescence microscopy to analyze in yeast cells the localization of 149 putative PM proteins that were C-terminally GFP-tagged at their native chromosomal locus (*Saier et al., 2016*; *Babu et al., 2012*; *Breker et al., 2014*; *Huh et al., 2003*). This collection included 16 different amino acid transporters (AATs) out of the 21 AATs that localize to the PM (*Bianchi et al., 2019*). In cells growing exponentially under defined (rich) conditions, we detected 50 GFP-tagged proteins at the PM, including eight different AATs and six different carbohydrate transporters (*Figure 1A*, *Figure 1—figure supplement 1A*, *Supplementary file 1*). A fraction of these proteins was additionally detected inside the vacuole (*Figure 1B*, *Figure 1—figure supplement 1A*), suggesting continuous turnover.

Others and we had shown earlier that amino acid and nitrogen starvation (hereafter starvation) induced the degradation of membrane proteins via the MVB pathway (*Müller et al., 2015*; *Jones et al., 2012*). Consistently, in response to starvation 15 (out of 50) different PM proteins were selectively removed and transported into the vacuole, including four AATs (Mup1, Can1, Lyp1, Tat2) and four carbohydrate transporters (Hxt1, Hxt2, Hxt3, Itr1) (*Figure 1B*, *Supplementary file 1*). The general amino acid permease Gap1 and the ammonium permease Mep2 were up-regulated and now localized to the PM. The localization of 35 GFP-tagged proteins to the PM remained largely unchanged, although in some instances the vacuolar GFP fluorescence was increased (*Figure 1A,B*, *Figure 1—figure supplement 1A*, *Supplementary file 1*). For five AATs, we did not detect signals under rich or starvation conditions (*Supplementary file 1*).

Next, we examined how amino acid availability could regulate AAT endocytosis. Therefore, we used the high-affinity methionine transporter Mup1 as a model cargo because its regulation in response to nutrient excess is well characterized (*Busto et al., 2018*; *Lee et al., 2019*; *Gournas et al., 2018*; *Guiney et al., 2016*; *Baile et al., 2019*). Moreover, Mup1 is one of the most abundant PM proteins and it is easy to follow its endocytosis and subsequent transport into the vacuole (*Busto et al., 2018*). In absence of methionine Mup1-GFP localized to the PM (*Figure 1—figure supplement 2A*). Low levels of methionine in the growth medium did not efficiently trigger its endocytosis (*Figure 1—figure supplement 2A,B*). Yet, above a critical methionine concentration the vast majority of Mup1-GFP was removed from the PM by endocytosis and was subsequently transported to the vacuole via the multivesicular body (MVB) pathway (*Figure 1—figure supplement 2A,B*). This was confirmed by western blot (WB) analysis from total cell lysates. Without methionine or with low levels of methionine in the growth medium only full-length Mup1-GFP was detected (*Figure 1—figure supplement 2B*, lanes 1+2). Once the critical methionine concentration was surpassed, Mup1-GFP was delivered into the vacuole. The proteolytic degradation of Mup1-GFP inside the vacuole then released free GFP, which remained stable and could be monitored by western blotting (*Figure 1—figure supplement 2B*, lane 4). Substrate-induced endocytosis was rapid and efficient: within 60–90 min after methionine addition Mup1 was almost quantitatively delivered to the vacuole. Excess methionine appeared to exclusively induce endocytosis of Mup1 since all other tested PM proteins remained at the cell surface (*Figure 1A*, *Supplementary file 1*). This exclusive substrate-induced endocytosis of Mup1 was also dependent on its methionine transport activity as shown by using the Mup1-G78N mutant, which cannot transition to the open-inward conformation

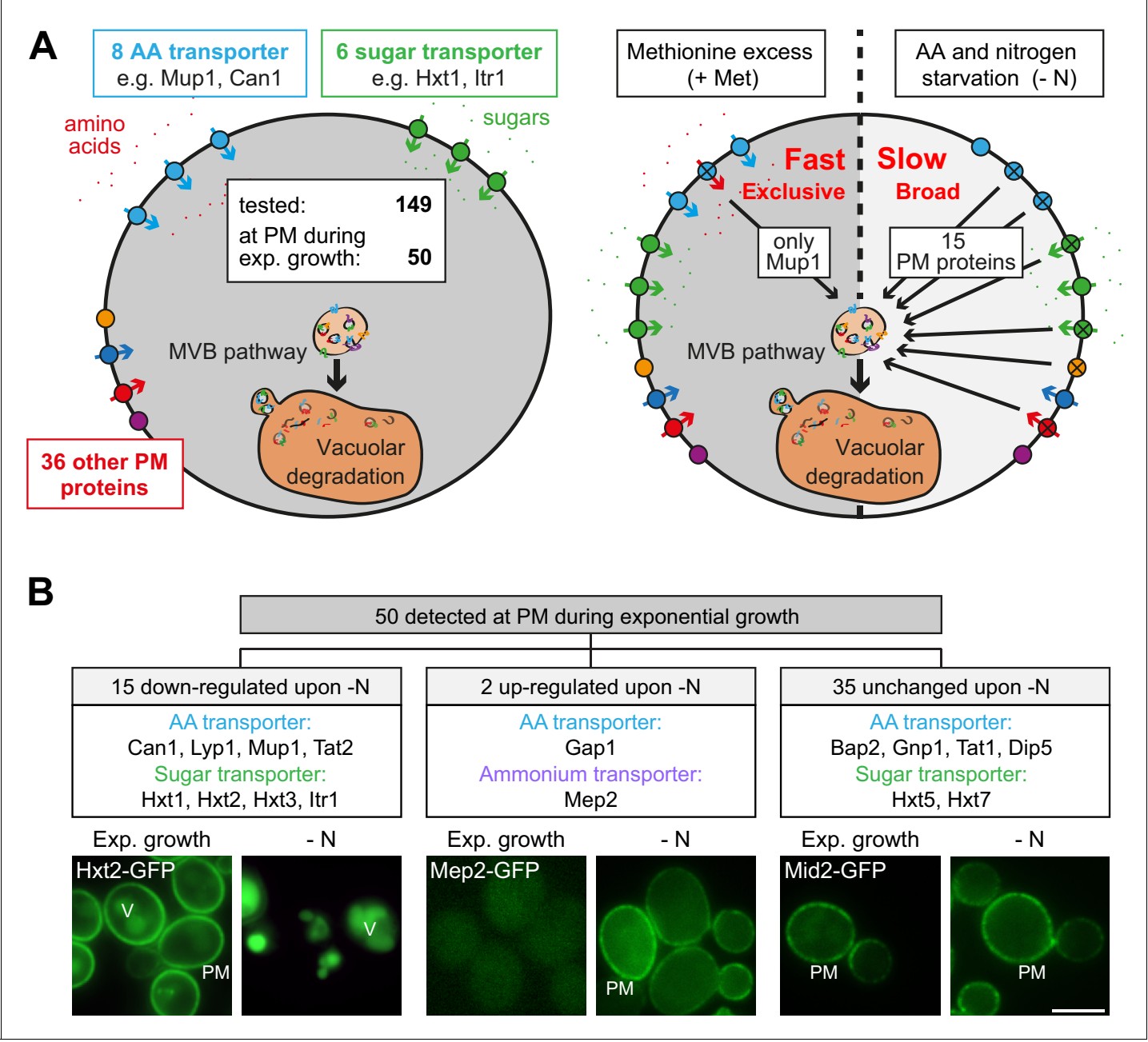

**Figure 1.** Amino acid and nitrogen starvation triggers broad but specific endocytosis and lysosomal degradation of plasma membrane proteins. (A) Left: a library of 149 yeast strains expressing chromosomally GFP-tagged membrane proteins was tested for plasma membrane (PM) localization during nutrient replete exponential growth. Right: verified PM proteins were starved 6–8 hr for amino acids and nitrogen (- N) or treated with 20 µg/ml L-methionine (+Met) after 24 hr of exponential growth. The localization of GFP was assayed by fluorescence microscopy. (B) Summary of the phenotypes of GFP-tagged PM proteins during starvation. Indicated are numbers of PM proteins that are down-regulated, up-regulated or unchanged compared to the exponential growth phase, each exemplified by one representative strain. PM: plasma membrane; V: vacuole. Scale bars = 5 µm. See also *Figure 1—figure supplements 1* and *2* and *Supplementary file 1*.

The online version of this article includes the following figure supplement(s) for figure 1:

**Figure supplement 1.** Localization of PM proteins during exponential, rich growth and starvation.

**Figure supplement 2.** Characterization of starvation- and substrate-induced endocytosis of Mup1.

(*Busto et al., 2018*; *Figure 1—figure supplement 2C*). These results are consistent with earlier reports showing that the high-affinity AATs Mup1, Can1, Tat2 and Lyp1 undergo exclusive endocytic down-regulation in response to excess of their respective substrates but remained stable at the PM when the substrate of another permease was present in excess (*Gournas et al., 2017*; *Busto et al., 2018*; *Nikko and Pelham, 2009*).

In contrast, the starvation-induced endocytosis of Mup1 was independent of its capability to transport its substrate (*Figure 1—figure supplement 2C*) and was also observed upon starvation for individual essential amino acids (*Figure 1—figure supplement 2D*). We re-evaluated the starvation-induced endocytosis of Mup1, Can1, Lyp1, Tat2, Ina1, Fur4, Hxt1, Hxt2 and Hxt3 in a different genetic background (SEY6210; *Figure 1—figure supplement 2A*, Figure 3C, *Figure 3—figure supplement 1C,D*, *Figure 6—figure supplement 2A*). The overall response was similar.

Hence, in response to starvation, a broad range of nutrient transporters, including four AATs (Mup1, Can1, Lyp1, Tat2) and four carbohydrate transporters (Hxt1, Hxt2, Hxt3, Itr1) as well as other membrane proteins, were selectively removed from the PM by endocytosis and transported into the vacuole within 3–6 hr. In contrast, substrate-induced endocytosis of AATs was quick and exclusive (*Figure 1A*). Earlier work showed that substrate-induced endocytosis of the AATs Mup1, Can1 and Lyp1 required TORC1 signaling to allow ubiquitination of the transporters by the HECT-type ubiquitin ligase Rsp5 in complex with the α-arrestin Art1 (*Lin et al., 2008*; *Guiney et al., 2016*; *Busto et al., 2018*; *Gournas et al., 2018*; *MacGurn et al., 2011*). Remarkably, the same set of AATs underwent simultaneous starvation-induced endocytosis. The molecular mechanism of their starvation-induced endocytosis was not clear.

## A genome-wide screen identifies regulators of starvation-induced endocytosis of AATs

To identify the genes that are required for the starvation-induced endocytosis of Mup1, we conducted a fluorescence-based genome-wide screen. We measured Mup1 sorting into the lumen of vacuoles via the MVB pathway by fusing the pH-sensitive GFP variant pHluorin to the C-terminus of Mup1 (Mup1-pHluorin) (*Henne et al., 2015*; *Prosser et al., 2010*). Fluorescence microscopy and flow cytometry (*Figure 2A,B*) showed a strong signal of Mup1-pHluorin at the PM of cells growing under defined rich conditions. In response to starvation, the fluorescence of Mup1-pHluorin was efficiently quenched in the acidic vacuoles of wild type (WT) cells, but not when the MVB pathway was blocked (e.g. ESCRT-I mutant *vps23Δ*) (*Figure 2A,B*). Hence Mup1-pHluorin is a suitable reporter to identify mutants that block trafficking of Mup1 from the PM into the vacuole.

Next, Mup1-pHluorin was introduced into the yeast non-essential knock-out collection. 4745 mutants expressing Mup1-pHluorin were grown in selection medium to exponential phase before they were subjected to starvation for 18–22 hr (*Figure 2C*). Automated flow cytometry was used to measure Mup1-pHluorin fluorescence intensity of at least 15,000 cells before and after starvation. The vast majority of mutants efficiently quenched the fluorescence of Mup1-pHluorin after starvation, but 128 mutants still exhibited Mup1-pHluorin fluorescence after starvation (*Figure 2C*, *Supplementary file 2*). To determine at which step Mup1-pHluorin transport into vacuoles was blocked in these mutants, we used live cell fluorescence microscopy. This allowed us to classify four phenotypes (*Figure 2D*, *Supplementary file 2*). Class one mutants retained Mup1-pHluorin at the PM after starvation (105 mutants, e.g. *apm2Δ* mutants). In class two mutants, Mup1-pHluorin was detected on small intracellular objects (seven mutants, e.g. *vam3Δ* mutants). Class three mutants accumulated Mup1-pHluorin in larger class E compartment-like objects (15 mutants, e.g. *vps4Δ* mutants). One Class four mutant (*pep4Δ*) did not efficiently quench Mup1-pHluorin fluorescence inside the vacuole, as it is deficient in the main lysosomal protease. Gene ontology (GO) analysis confirmed that our screen identified major general regulators of endosomal transport (e.g. the ESCRT complexes) and was enriched for proteins that are annotated as components of the PM, endosomes and vacuoles (*Figure 2—figure supplement 1*, *Supplementary file 3*) suggesting that it was successful. In addition, we identified many genes and components of protein complexes that would not be predicted to be involved in the regulation of endocytosis (*Figure 2—figure supplement 1*, *Supplementary file 2, 3*).

Next, we aimed to narrow down genes that function specifically during starvation-induced endocytosis and to distinguish them from general endocytic regulators (i.e. core components of endocytic trafficking machineries). Therefore, most mutants (124) were exposed to methionine excess for 90

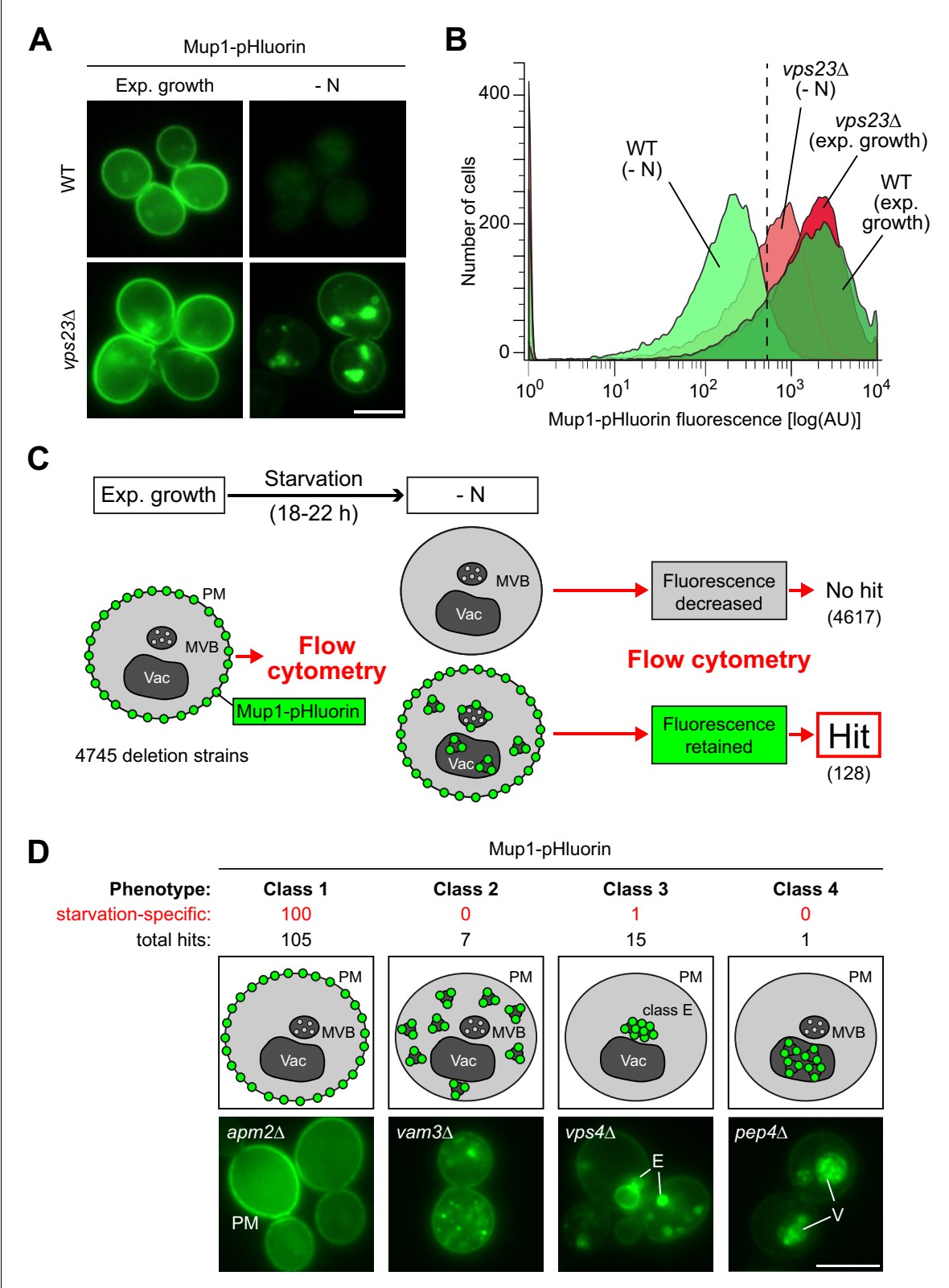

**Figure 2.** A genome wide screen revealed genes affecting Mup1-pHluorin endocytosis during starvation. (**A**) Live-cell fluorescence microscopy analysis of WT (BY4742) and *vps23Δ* cells expressing *MUP1-pHluorin* from plasmid and starved (- N) for 18–22 hr. The images exemplify quenched pHluorin fluorescence in vacuoles of wild type (WT)-like cells and retained fluorescence in mutants with defects in the starvation-induced endocytosis of Mup1-pHluorin. (**B**) The strains from (A) were exponentially grown in 96-well plates for 5 hr and starved (- N) for 18–22 hr. At least 15,000 cells from each strain
*Figure 2 continued on next page*

Figure 2 continued

and condition were analyzed by flow cytometry. The exemplified histograms display decrease of fluorescence in wild type (WT)-like strains and fluorescence retention in mutants with defects in the starvation-induced endocytosis of Mup1-pHluorin (e.g. *vps23Δ*). (**C**) Workflow of the flow-cytometry-based genome-wide screen for mutants defective in starvation-induced endocytosis of Mup1-pHluorin. (**D**) Summary of phenotypes of all mutants scored in the starvation-induced endocytosis screen (C) as determined by fluorescence microscopy. Class one mutants retain Mup1-pHluorin fluorescence at the plasma membrane (PM); class two mutants in small cytosolic objects; class three mutants in class E-like objects (E); class four mutants within vacuoles (V). Each phenotype is exemplified by one representative deletion mutant. Indicated are the numbers of strains that share a similar phenotype and the number of hits specific for starvation-induced endocytosis of Mup1-pHluorin (red). Scale bars = 5 μm. See also *Figure 2— figure supplement 1* and *Supplementary file 2*.

The online version of this article includes the following figure supplement(s) for figure 2:

**Figure supplement 1.** Gene ontology term analysis of the genome wide screen for genes affecting starvation-induced endocytosis of Mup1-pHluorin.

min and then subjected to live cell fluorescence microscopy. Of the mutants falling into classes 2–4, the majority was required for Mup1 sorting into the vacuole in response to both starvation and methionine excess. Among them were many mutants in subunits of well-characterized protein complexes that constituted the core machinery of endo-lysosomal trafficking (*Figure 2—figure supplement 1*, *Supplementary file 2*, *3*). However, the majority (100) of the class one mutants (Mup1-pHluorin retained at the PM) was specifically required for starvation-induced endocytosis but not for methionine-induced endocytosis (*Figure 2D*, *Supplementary file 2*).

## The α−arrestin Art2 is specifically required for amino acid transporter endocytosis in response to starvation

A key step for the endocytic down-regulation of PM proteins is their selective ubiquitination. This selectivity is mediated by α−arrestin molecules that direct the HECT-type ubiquitin ligase Rsp5 to ubiquitinate specific PM proteins (*Babst, 2020*). Among the 14 α−arrestins encoded by the yeast genome, our screen identified only Ecm21/Art2 to be specifically required for starvation-induced endocytosis of Mup1.

We compared Art2-dependent starvation-induced endocytosis to Art1-dependent substrate-induced endocytosis of Mup1-GFP. Efficient methionine-induced endocytosis of Mup1 required Art1 but not Art2 (*Figure 3A*), consistent with earlier observations (*Guiney et al., 2016*; *MacGurn et al., 2011*; *Busto et al., 2018*). Starvation-induced endocytosis of Mup1 was slower than methionine-induced endocytosis. In WT cells, 240–360 min after onset of starvation the majority of Mup1-GFP was removed from the PM and sorted into vacuoles for degradation (*Figure 3A,B*, *Figure 3—figure supplement 1A*). In *art2Δ* mutants, but not in *art1Δ* mutants, the vast majority of Mup1-GFP remained at the PM and was no longer delivered to vacuoles (*Figure 3A*). This was confirmed by WB analysis. In WT cells, the levels of full length Mup1-GFP decreased, while free GFP accumulated over time (*Figure 3B*, lane 2–4; *Figure 3—figure supplement 1A*), whereas in *art2Δ* mutants the majority of Mup1-GFP remained stable and only little free GFP could be detected (*Figure 3B* lane 7–8; *Figure 3—figure supplement 1A*). Re-expression of Art1 or Art2 in the corresponding mutants restored methionine- or starvation-induced Mup1 endocytosis, respectively (*Figure 3—figure supplement 1B*). Importantly, Art2 was required for the efficient starvation-induced endocytosis of all four AATs, Mup1, Can1, Lyp1, and Tat2 (*Figure 3A,C*). Interestingly, the same four AATs (Mup1, Can1, Lyp1 and Tat2) also depend on Art1 for efficient substrate-induced endocytosis (*Gournas et al., 2017*; *MacGurn et al., 2011*; *Lin et al., 2008*; *Nikko and Pelham, 2009*). Starvation-induced endocytosis of Ina1 was also dependent on Art2 (*Figure 3—figure supplement 1C*), while other PM proteins (e.g. the uracil transporter Fur4 or the hexose transporters Hxt1 and Hxt2) were largely independent of Art2 (*Figure 3—figure supplement 1D*).

These results suggested a swap between the α−arrestins Art1 and Art2 by amino acid availability as a rule for the regulation of four AATs. The starvation-induced endocytosis of Mup1, Can1, Lyp1 and Tat2 required Art2. Substrate-induced endocytosis of Mup1, Can1 and Lyp1 required Art1, whereas in case of Tat2 substrate-induced endocytosis was less stringent and could be mediated either by Art1 or Art2 (*Nikko and Pelham, 2009*).

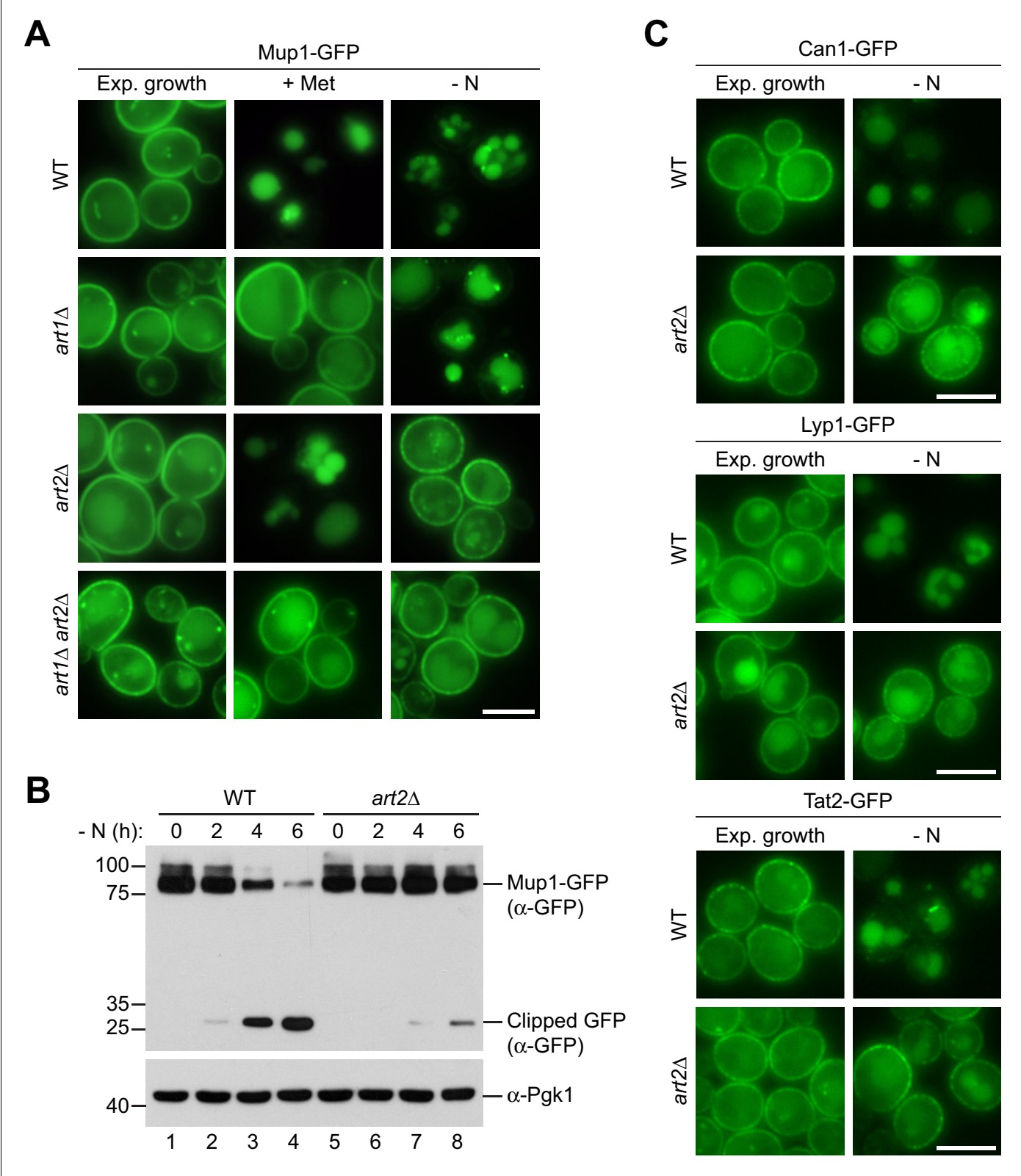

**Figure 3.** Art1 and Art2 are non-redundant in promoting substrate- and starvation-induced endocytosis of amino acid transporters. (**A**) Live-cell fluorescence microscopy analysis of Mup1-GFP endocytosis in wild type (WT), *art1Δ*, *art2Δ* and *art1Δ art2Δ* cells expressing *MUP1-GFP* from plasmid. Cells were treated with 20 μg/ml L-methionine (+ Met) for 1.5 hr or starved (- N) for 6 hr after 24 hr exponential growth. (**B**) SDS PAGE and western blot analysis with the indicated antibodies of whole cell protein extracts from wild type (WT) and *art2Δ* cells expressing *MUP1-GFP* that were starved (- N) for

*Figure 3 continued on next page*

*Figure 3 continued*

the indicated times after 24 hr exponential growth. Quantification in *Figure 3—figure supplement 1A*. (C) Live-cell fluorescence microscopy analysis of wild type (WT) and *art2Δ* cells expressing *CAN1-GFP*, *LYP1-GFP* or *TAT2-GFP*. Cells were starved (- N) for 6 hr after 24 hr exponential growth. Scale bars = 5 μm. See also *Figure 3—figure supplement 1*.

The online version of this article includes the following figure supplement(s) for figure 3:

**Figure supplement 1.** Art2-dependent and -independent starvation-induced endocytosis of PM proteins.

## Up-regulation of Art2 by the GAAC pathway drives starvation-induced endocytosis of Mup1

Our experiments so far demonstrated that Art1 and Art2 mediate the endocytic down-regulation of Mup1, Can1 and Lyp1 in response to changes in amino acid availability in a mutually exclusive manner. Hence, their activity must be strictly regulated. α−arrestin proteins are subject to complex multi-level regulation and their activation can require transcriptional regulation, (de-)phosphorylation and ubiquitination (*Hovsepian et al., 2017*; *Lin et al., 2008*; *Becuwe et al., 2012b*). WB analysis of total cell lysates indicated that 6xHis-TEV-3xFlag-tagged Art1 protein levels were unchanged in response to starvation (*Figure 4A*, lanes 2,3), whereas the protein levels of the functional 170kD protein Art2-HTF (*Figure 4—figure supplement 1A*) were up-regulated in response to starvation (*Figure 4A*, lanes 5,6; *Figure 4—figure supplement 1B*). Quantitative reverse transcription (RT) PCR analysis indicated that also the *ART2* mRNA levels increased during starvation (*Figure 4B*).

In search of the pathway responsible for induction of Art2, we queried the results of our screen and identified the eIF2 kinase Gcn2 and several more *GCN* genes encoding key components of the general amino acid control (GAAC) pathway (*Supplementary file 2*). Gcn2 is activated during amino acid starvation by unloaded tRNAs and subsequently phosphorylates eIF2α to reduce protein synthesis in general, but thereby stimulates specifically the translation of the transcription factor Gcn4. Gcn4 then activates the transcription of genes, many of which are involved in amino acid metabolism (*Hinnebusch, 2005*). *gcn4Δ* was not scored in our genome-wide screen because it failed the quality control due to its slow growth phenotype. Interestingly, the promoter of *ART2* contained predicted Gcn4-binding sites (*Venters et al., 2011*; *Schuldiner et al., 1998*). Disrupting the GAAC pathway (*gcn4Δ*) eliminated the induction of Art2 in response to starvation at mRNA and protein levels (*Figure 4B,C*, *Figure 4—figure supplement 1C*). Consistently, the starvation-induced endocytosis of Mup1-GFP was hampered in *gcn4Δ* cells and several other *gcn* mutants (*Figure 4D*, *Figure 4—figure supplement 1D*). Also, starvation-induced endocytosis of Can1 was dependent on the GAAC pathway (*Figure 4—figure supplement 1E*). When we introduced mutations in the predicted Gcn4-binding sites in the *ART2* promoter, Art2 protein levels no longer increased in response to starvation, and starvation-induced endocytosis of Mup1-GFP was impaired (*Figure 4E*, *Figure 4—figure supplement 1F*). When the *ART2* promoter region was used to replace the *ART1* promoter, it also induced the expression of Art1 during starvation (*Figure 4—figure supplement 2A*). Yet, the up-regulation of Art1 protein levels driven by the *ART2* promoter failed to restore starvation-induced endocytosis of Mup1-GFP in *art2Δ* cells (*Figure 4—figure supplement 2B*). This construct was functional for substrate-induced endocytosis of Mup1-GFP in *art1Δ* cells (*Figure 4—figure supplement 2B*).

The expression of a constitutively translated Gcn4[C] construct (*Mueller and Hinnebusch, 1986*) increased Art2 protein levels already under rich conditions, as revealed by WB analysis (*Figure 4F*, compare Art2 protein levels in lanes 1 and 3, *Figure 4—figure supplement 1G*), and drove unscheduled Mup1-GFP endocytosis (*Figure 4F*, lower panel). Consistently, over-expression of Art2 in WT cells or in *gcn4Δ* mutants using the strong and constitutively active *TDH3* promoter (*Figure 4—figure supplement 2C*) initiated Mup1-GFP endocytosis already under nutrient replete conditions (*Figure 4G*) and bypassed the requirement of Gcn4 during starvation.

These results indicated that during amino acid starvation, Art2 transcription was induced via the GAAC pathway, leading to the up-regulation of Art2 protein levels. Moreover, it seemed that the up-regulation of Art2 protein levels was sufficient to drive Mup1 endocytosis. Hence, under amino acid replete conditions Art2 must be tightly repressed to prevent AAT endocytosis.

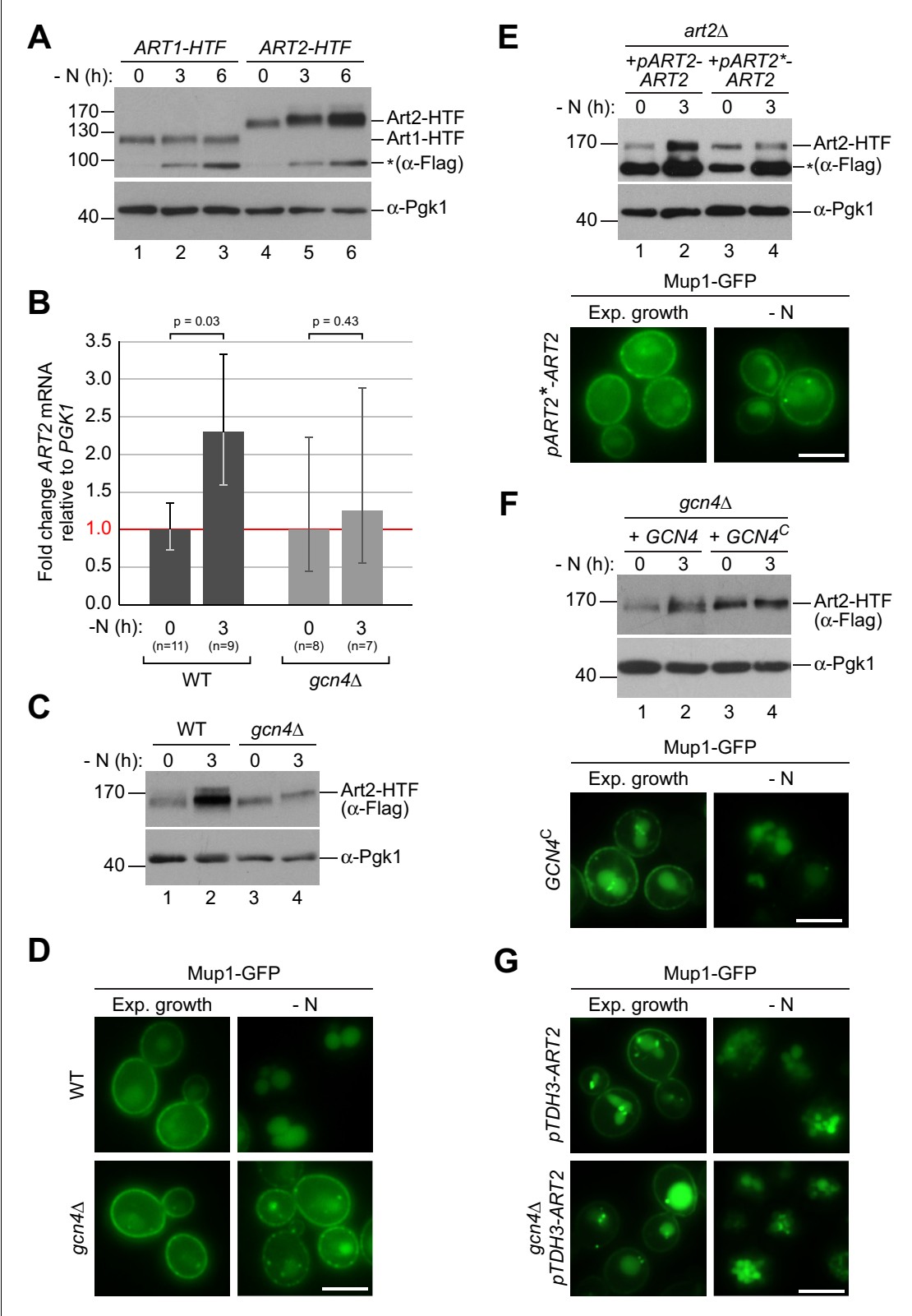

**Figure 4.** The general amino acid control pathway promotes starvation-induced endocytosis Mup1 by up-regulating Art2. (A) SDS PAGE and western blot analysis with the indicated antibodies of whole cell protein extracts from WT cells expressing *ART1-HTF* or *ART2-HTF*. Cells were starved (- N) for the indicated times after 24 hr exponential growth. The asterisk indicates a non-specific background band of the FLAG antibody. Quantification in *Figure 4—figure supplement 1B*. (B) RT-qPCR analysis of *ART2* transcript levels (normalized to the stable *PGK1* transcript) in wild type (WT) and *gcn4Δ*

*Figure 4 continued on next page*

Figure 4 continued

cells. Cells were starved (- N) for 3 hr after 24 hr exponential growth. Values are presented as fold-change of the starting values (t = 0). Error bars represent the standard deviation. Statistical significance was assessed by Student's t-test. (C) SDS PAGE and western blot analysis with the indicated antibodies of whole cell protein extracts from the indicated strains expressing *ART2-HTF*. Cells were starved (- N) for 3 hr after 24 hr exponential growth. Quantification in *Figure 4—figure supplement 1C*. (D) Live-cell fluorescence microscopy analysis of the indicated strains expressing *MUP1-GFP* from plasmid. Cells were starved (- N) for 6 hr after 24 hr exponential growth. (E), (F) The indicated strains were analyzed as in C) (upper panels) and D) (lower panels). Quantification of western blots in *Figure 4—figure supplement 1F,G*. (G) Live-cell fluorescence microscopy analysis of *art2Δ* or *gcn4Δ* cells expressing pRS415-*MUP1-GFP* and pRS416-*pTDH3-ART2* starved (- N) for 6 hr after 24 hr exponential growth. Scale bars = 5 μm. See also *Figure 4—figure supplements 1* and *2*.

The online version of this article includes the following figure supplement(s) for figure 4:

**Figure supplement 1.** The general amino acid control pathway promotes starvation-induced endocytosis Mup1 by up-regulating Art2 – supporting experiments and quantifications.

**Figure supplement 2.** Upregulation of Art1 cannot substitute Art2 in starvation-induced endocytosis of Mup1.

## Art2 directs Rsp5-dependent ubiquitination of C-terminal lysine residues in Mup1

To determine how Art2 contributed to Mup1 endocytosis, we examined its role in Mup1 ubiquitination. WT cells were harvested and Mup1-GFP was immunoprecipitated in denaturing conditions before and at different time points after starvation. Equal amounts of immunoprecipitated full-length Mup1-GFP were subjected to SDS-PAGE and WB analysis to compare the extent of its ubiquitination at different time points (*Figure 5A*). This analysis indicated that a pool of Mup1 was ubiquitinated prior to the onset of starvation (*Figure 5A*, lane 1). At the onset of starvation, ubiquitination of Mup1-GFP appeared to decrease for some time (*Figure 5A* lanes 2–4), until ubiquitination of Mup1-GFP began to increase again after 2–3 hr during starvation (*Figure 5A* lanes 4–6), temporally coinciding with Art2 induction and starvation-induced endocytosis. Mup1-GFP was still ubiquitinated in *art2Δ* cells growing under rich conditions (*Figure 5B* lane 4) and seemingly de-ubiquitinated at the onset of starvation, but the increase of ubiquitination during starvation was no longer observed (*Figure 5B*, lane 6). Hence, Art2 was essential for the starvation-induced ubiquitination of Mup1.

α−Arrestins use PY motifs to bind to at least one of the three WW domains of Rsp5 (*Lin et al., 2008*). Starvation-induced endocytosis of Mup1 (but not methionine-induced endocytosis) was particularly dependent on the WW3 domain of Rsp5 (*Figure 5C,D*, *Figure 5—figure supplement 1A*). Art2 has four putative PY motifs (*Figure 5C*). Mutations in the PY motif that resides within the predicted arrestin fold of Art2 (P748,P749,Y750) reduced starvation-induced endocytosis of Mup1-GFP (*Figure 5E*). The Art2$^{P748A,P749A,Y750A}$ mutant was expressed at similar levels than the WT protein (*Figure 5—figure supplement 1B*). We suggest that interaction between WW3 in Rsp5 and the PY motif (748-750) of Art2 was required for the efficient starvation-induced endocytosis of Mup1.

To identify lysine residues in Mup1 that were ubiquitinated in response to starvation, Mup1-GFP was immunoprecipitated before and 3 hr after starvation, digested and subjected to liquid chromatography-mass-spectrometry (LC-MS) (*Figure 5—figure supplement 1C*). Two N-terminal lysine residues (K16 and K27) and two C-terminal lysine residues (K567 and K572) in Mup1 were found to be ubiquitinated (*Figure 5F*. *Figure 5—figure supplement 1C*), consistent with available high-throughput proteomic datasets (*Swaney et al., 2013*; *Iesmantavicius et al., 2014*). Earlier reports showed that two N-terminal lysine residues (K27, K28) in Mup1 were ubiquitinated by Art1-Rsp5 and required for methionine-induced endocytosis (*Guiney et al., 2016*; *Busto et al., 2018*; *Ghaddar et al., 2014*; *Figure 5F*). The role of ubiquitination at C-terminal lysine residues was not clear.

To determine how these N- and C-terminal lysine residues contributed to Mup1 ubiquitination under different growth conditions, we mutated the N-terminal (K27, K28) or C-terminal (K567, K572) lysine residues to arginine and additionally generated a quadruple mutant (K27,28,567,572R). Equal amounts of immunoprecipitated Mup1-GFP, Mup1$^{K27,28R}$-GFP, Mup1$^{K567,572R}$-GFP and Mup1$^{K27,28,567,572R}$-GFP from exponentially growing cells or after starvation or methionine treatment were subjected to SDS-PAGE and WB analysis to compare their ubiquitination (*Figure 5G*, *Figure 5—figure supplement 1D*). Preventing the ubiquitination of the C-terminal lysine residues (K567,572R) abrogated starvation-induced ubiquitination of Mup1 (*Figure 5G*, lanes 7–9), whereas Mup1$^{K27,28R}$-GFP was still ubiquitinated after starvation (*Figure 5G*, lanes 4–6). Upon methionine

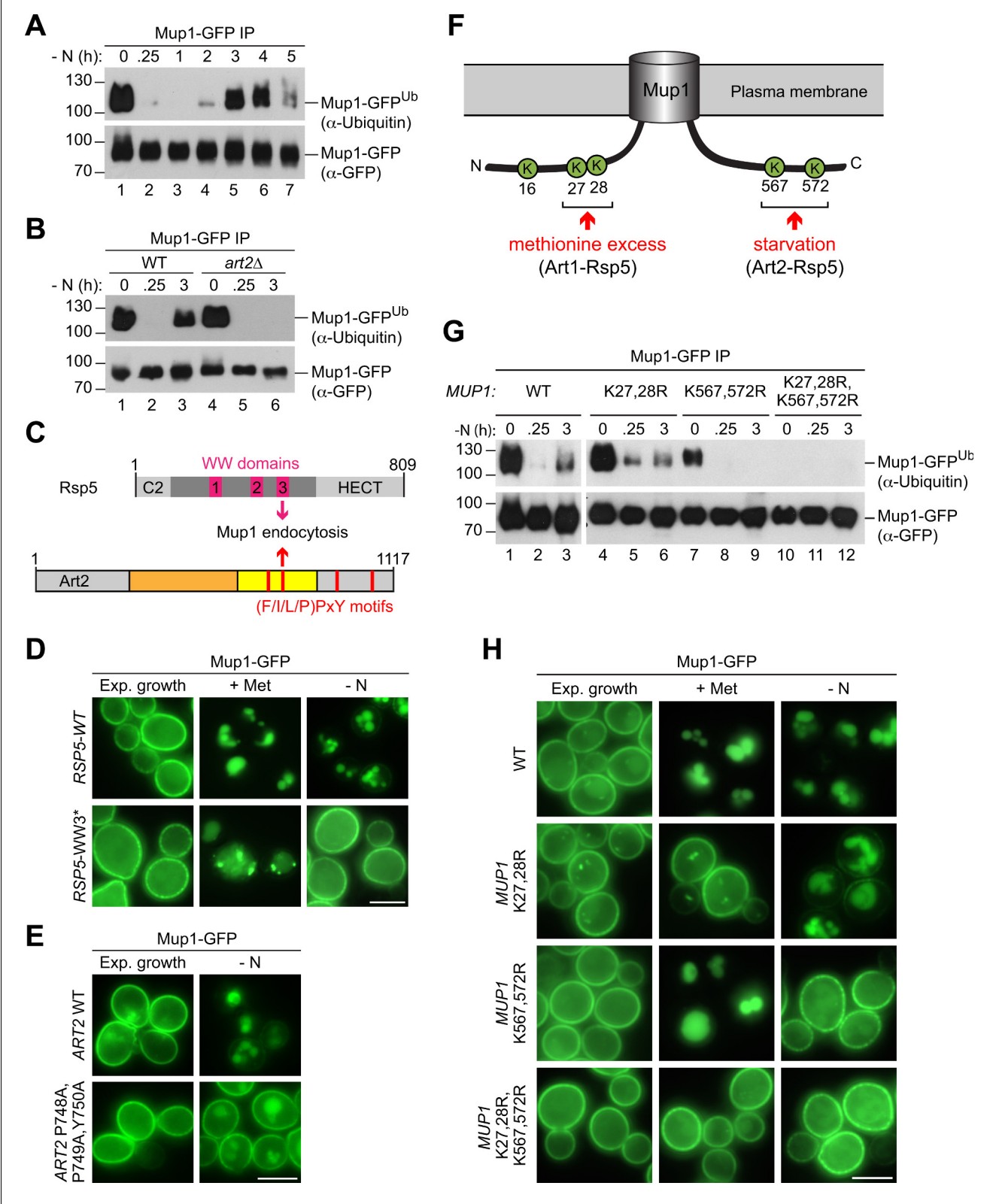

**Figure 5.** Art2-Rsp5 mediates the starvation-induced ubiquitination of Mup1-GFP at two specific C-terminal lysine residues. (**A**), (**B**) SDS PAGE and western blot analysis with the indicated antibodies of immunoprecipitated Mup1-GFP from WT cells or *art2Δ* cells starved for the indicated times after 24 hr of exponential growth. Equal amounts of immunoprecipitated Mup1-GFP were loaded to compare the extent of ubiquitination. (**C**) Scheme depicting the domain arrangement of Rsp5 and Art2, indicating the localization of the WW domains and PY motifs required for starvation-induced

*Figure 5 continued on next page*

Figure 5 continued

endocytosis of Mup1. (D) Live-cell fluorescence microscopy analysis of *rsp5Δ* cells expressing pRS416-*MUP1-GFP* and pRS415-*HTF-RSP5-WT* (wild type) or pRS415-*HTF-RSP5-WW3\**. Cells were treated with 20 µg/ml L-methionine (+ Met) for 1.5 hr or starved (- N) for 6 hr after 24 hr exponential growth. (E) Live-cell fluorescence microscopy analysis of *art2Δ* cells expressing *MUP1-GFP* and pRS416-*ART2* (WT) or pRS416-*ART2 P748A,P749A,Y750A*. Cells were starved (- N) for 6 hr after 24 hr exponential growth. (F) Scheme of Mup1 topology with the N- and C-terminal ubiquitination sites targeted during substrate excess by Art1-Rsp5 and during starvation by Art2-Rsp5. Ubiquitinated lysines (K) shown in green with numbers corresponding to amino acid positions in the Mup1 sequence. (G) WT cells expressing *MUP1-GFP* WT or the indicated *MUP1-GFP* mutants starved for the indicated times after 24 hr of exponential growth analyzed as in B). (H) Live-cell fluorescence microscopy analysis of cells expressing *MUP1-GFP* (wild type (WT)), *MUP1 K27,28R-GFP*, *MUP1 K567,572R-GFP* or *MUP1 K27,28,567,572R-GFP* as in D). Scale bars = 5 µm. See also *Figure 5—figure supplement 1*.

The online version of this article includes the following figure supplement(s) for figure 5:

**Figure supplement 1.** Post-translational modifications regulating starvation- and substrate-induced endocytosis of Mup1.

treatment, Mup1$^{K567,572R}$-GFP ubiquitination was comparable to WT Mup1 (*Figure 5—figure supplement 1D*, lanes 1–4), while the ubiquitination of Mup1$^{K27,28R}$ was impaired as reported previously (*Figure 5—figure supplement 1D*, lanes 5,6) (*Guiney et al., 2016*; *Busto et al., 2018*; *Ghaddar et al., 2014*). The quadruple mutant Mup1$^{K27,28,567,572R}$-GFP always appeared to be devoid of ubiquitination (*Figure 5G*, lanes 11,12; *Figure 5—figure supplement 1D*, lanes 7,8), suggesting that the two N-terminal and the two C-terminal lysine residues are the major sites for ubiquitination. Consistently, the quadruple lysine mutation rendered Mup1$^{K27,28,567,572R}$-GFP refractory to methionine- and starvation-induced endocytosis (*Figure 5H*). Mutations in the N-terminal lysine residues (K27,28R) blocked methionine-induced endocytosis, but not starvation-induced endocytosis. In contrast, starvation-induced endocytosis of Mup1$^{K567,572R}$-GFP was specifically blocked, but not its methionine-induced endocytosis (*Figure 5H*).

Collectively, these results suggested that Art2 directs Rsp5 to the ubiquitination of the C-terminal lysine residues K567 and K572 during starvation, whereas Art1-Rsp5 complexes ubiquitinated Mup1 on the two N-terminal lysine residues K27 and K28 in response to methionine excess (*Figure 5F*).

## The C-terminus of Mup1 contains an acidic sorting signal for Art2

In addition to ubiquitination, our mass-spectrometry analysis of immunoprecipitated Mup1-GFP from starved cells identified potential phosphorylation sites at the N-terminus (S9,23,31,42 and T34) and at the C-terminus of Mup1 (T552 and S568,573) (*Figure 6A*, *Figure 5—figure supplement 1C*). Additional phosphorylation on T560 was reported in other studies (*Swaney et al., 2013*; *Iesmantavicius et al., 2014*). Of note, the C-terminal phosphorylation sites were proximal (T552, T560) or directly adjacent (S568, S573) to lysine residues K567 and K572 that were ubiquitinated in an Art2-dependent manner in response to starvation (*Figure 6A*, *Figure 5—figure supplement 1C*).

Mutation of the C-terminal threonine residues to alanine (T552,560A) to prevent phosphorylation specifically blocked starvation-induced endocytosis of Mup1$^{T552,560A}$-GFP, but not methionine-induced endocytosis (*Figure 5—figure supplement 1E*). Mutation of these phosphorylation sites also prevented ubiquitination of Mup1 during starvation (*Figure 5—figure supplement 1F*). Blocking the phosphorylation of S568 and S573 also impaired starvation-induced endocytosis of Mup1$^{S568,573A}$-GFP (*Figure 5—figure supplement 1E*), yet starvation-induced ubiquitination could still be detected (*Figure 5—figure supplement 1F*). Mutation of most N-terminal serine and threonine residues, including all experimentally determined phosphorylation sites, to alanine (T6,25,26,34A,S9,22,23,24,31,33,42A) did not affect starvation-induced endocytosis of Mup1 (*Figure 5—figure supplement 1G*). It seemed that the phosphorylation of C-terminal threonine residues (T552, 560) contributed to the starvation-induced Art2-Rsp5 ubiquitination of Mup1 and thus might add an additional layer of regulation, similar to the recognition of GPCRs by β–arrestins in mammalian cells (*Shukla et al., 2014*; *Nikko et al., 2008*; *Marchal et al., 2000*; *Hicke et al., 1998*).

An extended acidic Art1 sorting motif at the N-terminus of Mup1 is required for ubiquitination of K27,28 and subsequent Mup1 endocytosis in response to methionine excess (*Guiney et al., 2016*; *Busto et al., 2018*). Consistent with our model involving two distinct mechanisms for Mup1 endocytosis, mutations in the acidic N-terminal Art1 sorting motif of Mup1 (D43A,G47A,Q49A,T52A,L54A) blocked Art1-Rsp5-dependent methionine-induced endocytosis, but not Art2-Rsp5-dependent starvation-induced endocytosis (*Figure 6—figure supplement 1A*).

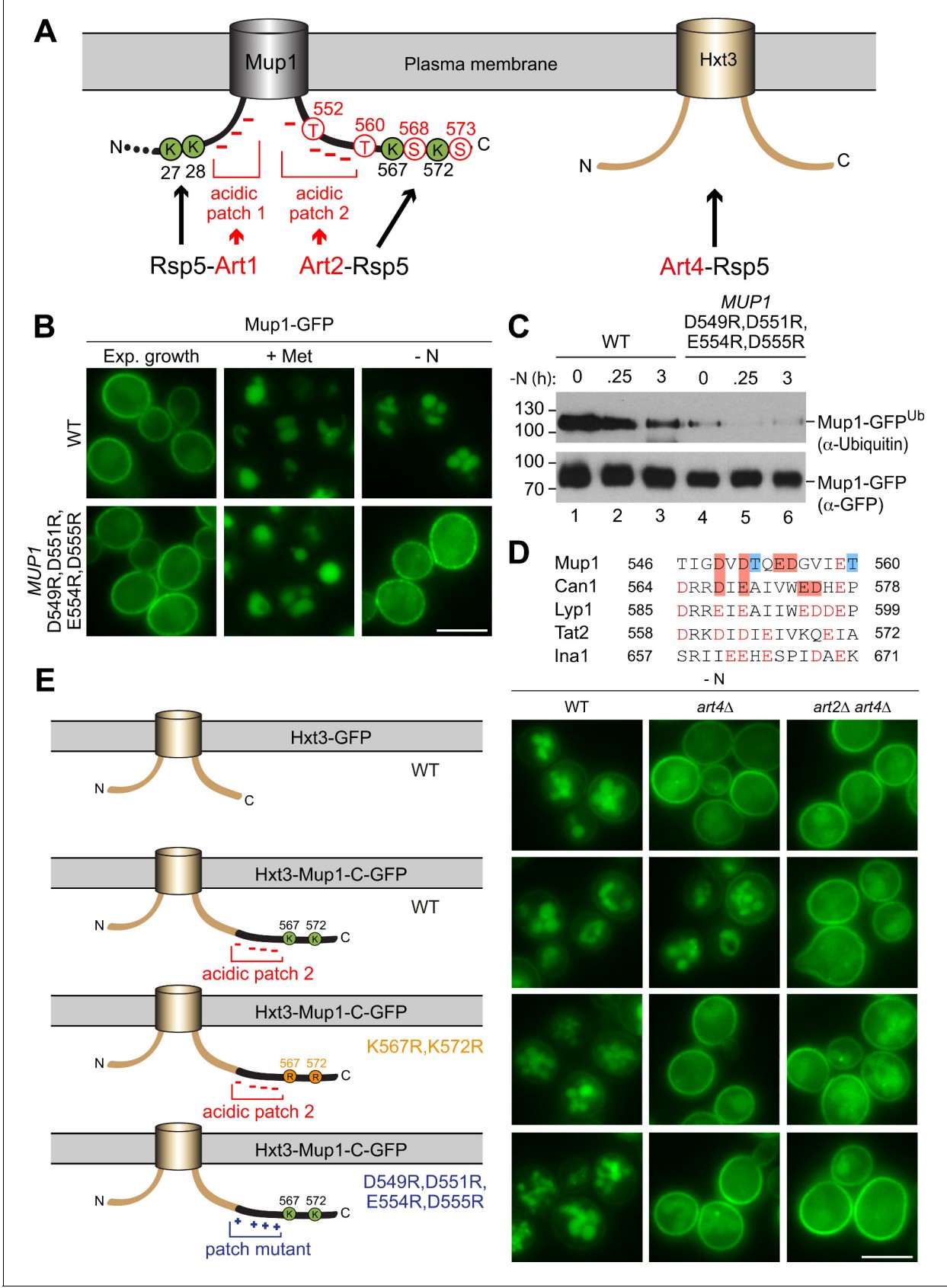

**Figure 6.** The C-terminus of Mup1 harbors a transplantable, starvation-responsive acidic degron. (A) Left: scheme of Mup1 topology with N- and C-terminal ubiquitination sites and acidic patches targeted by Art1-Rsp5 and Art2-Rsp5, respectively, and the C-terminal phosphorylation sites of Mup1 that promote its starvation-induced endocytosis. Ubiquitinated lysines (K) shown in green and phosphorylated serines (S) and threonines (T) in red with numbers corresponding to amino acid positions in the Mup1 sequence. Right: Hxt3 as an Art4-Rsp5 dependent cargo during nitrogen starvation. (B) Live-cell fluorescence microscopy analysis of Mup1-GFP endocytosis in cells expressing *MUP1-GFP* (wild type (WT)) or *MUP1 D549R,D551R,E554R, D555R-GFP*. Cells were treated with 20 µg/ml L-methionine (+ Met) for 1.5 hr or starved (- N) for 6 hr after 24 hr exponential growth. (C) SDS PAGE and western blot analysis with the indicated antibodies of immunoprecipitated Mup1-GFP from cells expressing *MUP1-GFP* (WT) or *MUP1 D549R,D551R, E554R,D555R-GFP* starved for the indicated times after 24 hr of exponential growth. Equal amounts of immunoprecipitated Mup1-GFP were loaded to compare the extent of ubiquitination. (D) Amino acid sequence alignment of the C-terminal acidic patches of Mup1, Can1, Lyp1, Tat2 and Ina1. The boxes indicate acidic residues (red) and phosphorylation sites (blue), which are required for Art2-dependent starvation-induced endocytosis. Red letters illustrate further acidic residues. (E) Live-cell fluorescence microscopy analysis of wild type (WT), *art4Δ* and *art2Δ art4Δ* cells expressing *HXT3-GFP* (top), *HXT3-MUP1-C-GFP* (second row), *HXT3-MUP1-C K567,572R-GFP* (third row) or *HXT3-MUP1-C D549R,D551R,E554R,D555R-GFP* (bottom). Cells were starved (- N) for 6 hr after 24 hr exponential growth. Scale bars = 5 µm. See also *Figure 6—figure supplements 1* and *2*.
The online version of this article includes the following figure supplement(s) for figure 6:

**Figure supplement 1.** The role of acidic patches in AATs for starvation-induced endocytosis.
**Figure supplement 2.** Art4-dependent starvation-induced endocytosis of Hxt3.

The C-terminus of Mup1 also contains an acidic patch (D549-D555), close to the ubiquitination sites K567 and K572 and the C-terminal threonine phosphorylation sites (T552, 560) involved in starvation-induced endocytosis (*Figure 6A,B*). Mutation of the acidic residues in this region to basic amino acids (R) demonstrated that this C-terminal acidic region was specifically required for starvation-induced endocytosis. Live cell fluorescence microscopy revealed that Mup1$^{D549R,D551R,E554R,D555R}$-GFP remained at the PM in response to starvation, whereas methionine-induced endocytosis was not impaired (*Figure 6B*). Even Art2 overexpression failed to induce endocytosis during exponential growth or starvation when the C-terminal acidic patch in Mup1 was mutated (*Figure 6—figure supplement 1B*). Moreover, immunoprecipitation of Mup1$^{D549R,D551R,E554R,D555R}$-GFP and subsequent SDS-PAGE and WB analysis revealed that it was no longer efficiently ubiquitinated (*Figure 6C*, lanes 4–6), suggesting that the C-terminal acidic patch was essential for the Art2-Rsp5-dependent ubiquitination during starvation.

Comparing the amino acid sequences of the C-terminal tails of the four Art2-dependent AATs (Mup1, Can1, Tat2 and Lyp1) and of Ina1 indicated similar acidic patches (*Figure 6D*). To analyze if the acidic patch in Can1 also contributed to starvation-induced endocytosis, we mutated D567, E569, E574 and E575 to arginine. Mutant Can1$^{D567R,E569R,E574R,D575R}$-GFP mostly localized to the PM under growing conditions. Importantly, the Art2-dependent starvation-induced down-regulation of Can1$^{D567R,E569R,E574R,D575R}$-GFP was impaired (*Figure 6—figure supplement 1C*). These results imply that Mup1, Can1 and potentially also Lyp1, Tat2 and Ina1 have acidic amino acid sequences at their C-termini that could serve as sorting signal for Art2-mediated starvation-induced endocytosis.

## The C-terminal acidic sorting signal of Mup1 is sufficient for Art2-dependent starvation-induced endocytosis

It seemed that the last 26 amino acid residues (aa 549–574) of Mup1 harbor three features that are collectively required specifically for starvation-induced endocytosis: putative phosphorylation sites, the acidic patch and the ubiquitination sites. Hence, we tested if the C-terminal region of Mup1 was sufficient to convert an Art2-independent cargo into an Art2 cargo. We selected the low-affinity glucose transporter Hxt3, which was efficiently removed from the PM in response to starvation (*Figure 6—figure supplement 2A*, *Figure 1B*, *Supplementary file 1*). Live cell fluorescence microscopy and WB analysis showed that starvation-induced endocytosis of Hxt3-GFP was independent of Art2, but instead required Art4 (*Figure 6E*, *Figure 6—figure supplement 2A*). In *art4Δ* mutants, but not in *art2Δ* mutants, Hxt3-GFP remained mostly at the PM (*Figure 6—figure supplement 2A*) in response to starvation, and its vacuolar degradation was impaired (*Figure 6—figure supplement 2B*, lane 4, *Figure 6—figure supplement 2C*). However, when the C-terminal 30 amino acids of Mup1 (aa 545–574) were fused onto the C-terminus of Hxt3 (*Figure 6E*), they restored starvation-induced endocytosis in *art4Δ* mutants. This Hxt3-Mup1-C-GFP chimera was now efficiently removed from the PM and transported to the vacuole in the vast majority of *art4Δ* cells (86%, n = 76 cells) (*Figure 6E*), most likely because it became an Art2 substrate. Indeed, the additional deletion of

*ART2* (*art2Δ art4Δ*) blocked starvation-induced endocytosis and demonstrated that Art2 now mediated the degradation of Hxt3-Mup1-C-GFP in *art4Δ* mutants. The degradation of Hxt3-Mup1-C-GFP and the concomitant accumulation of free GFP were further examined by SDS-PAGE and WB analysis from total cell extracts (*Figure 6—figure supplement 2D, E*). This analysis showed that the majority of the full-length Hxt3-Mup1-C-GFP chimera was degraded in WT cells, *art2Δ* and *art4Δ* mutants during starvation (*Figure 6—figure supplement 2D*, lanes 2, 4, 8), but only poorly in *art2Δ art4Δ* double mutants (*Figure 6—figure supplement 2D*, lane 6, *Figure 6—figure supplement 2E*).

The Art2-dependent endocytosis of the Hxt3-Mup1-C-GFP chimera required two key features provided by the C-terminus of Mup1 (the acidic patch and the two C-terminal lysine residues), since in *art4Δ* cells starvation-induced endocytosis of Hxt3-Mup1-C$^{K567,572R}$-GFP and Hxt3-Mup1-C$^{D549R, D551R,E554R,D555R}$-GFP was blocked (*Figure 6E*).

Taken together, these results demonstrate that the C-terminus of Mup1 (aa 545–574) encodes a portable acidic sorting signal that can be recognized by Art2 and directs Rsp5 to ubiquitinate specifically two proximal lysine residues to promote starvation-induced endocytosis.

## A basic patch of Art2 is required for starvation-induced degradation of Mup1

After having defined that the C-terminus of Mup1 (and possibly also the C-terminus of further Art2-targets) provides a degron sequence for Art2-Rsp5 complexes, we addressed how it could be recognized. Upon inspection of the predicted arrestin domain in Art2, we noted a stretch of positively charged residues within the arrestin-C domain (*Figure 7A*). Converting these basic residues into an acidic patch (Art2$^{K664D,R665D,R666D,K667D}$) abolished starvation-induced endocytosis of Mup1 (*Figure 7B*). Western blot analysis of total cell lysates showed that the Art2 basic patch mutant protein was expressed at similar levels as WT Art2 and was also upregulated after 3 hr of starvation (*Figure 7—figure supplement 1A*, lane 6). In addition, the Art2 basic patch mutant also impaired, at least partially, starvation-induced endocytosis of Can1, Lyp1 and Ina1, while the endocytosis of Tat2 was independent of the basic patch (*Figure 7B*, *Figure 7—figure supplement 1B,C*).

Overall, it seemed that Art2 employed a positively charged region in its arrestin fold to recognize C-terminal acidic patches in at least three different AATs (Mup1, Can1, Lyp1) and thus mediate their endocytosis in starvation conditions.

## Discussion

We have made progress toward understanding how yeast cells selectively re-configure their repertoire of transporters at the PM in response to their nutritional status. The model in *Figure 7C* provides the conceptual framework for the mutually exclusive but complementary action of Art1-Rsp5 and Art2-Rsp5 ubiquitin ligase complexes for mediating the selective endocytosis of the methionine transporter Mup1 in response to changes in amino acid availability. Similar concepts may apply for the arginine transporter Can1, the lysine transporter Lyp1 and the tryptophan and tyrosine transporter Tat2. Based on our results and work from others, the endocytic down-regulation of these AATs can be described by the following set of rules: (1) Reciprocal (de-)activation of the α−arrestins Art1 and Art2. (2) Art1 or Art2 each recognize different acidic sorting signals on their client AATs that may require additional phosphorylation. (3) To do so, they employ basic patches in their extended arrestin C-domains and defined PY motifs to orient Rsp5 with high specificity toward proximal lysine residues. These rules satisfy the plasticity required for different α−arrestin and AAT interactions that drive exclusive or relatively broad substrate specificity depending on the metabolic context.

While both Art1 and Art2 lead to the degradation of AATs, they answer to distinct metabolic cues and are thus wired into distinct signaling pathways. Activation of Art1 by amino acid influx requires the coordinated interplay of TORC1 signaling to inactivate Npr1 (a kinase that negatively regulates Art1) and the action of phosphatases (*Gournas et al., 2017*; *Lee et al., 2019*; *MacGurn et al., 2011*; *Tumolo et al., 2020*). In response to amino acid limitation, TORC1 is no longer active. This will activate Npr1 to phosphorylate Art1, thereby inactivating it. At the same time, the lack of amino acids will activate the eIF2α kinase Gcn2. Gcn2 will phosphorylate eIF2α, which leads to the global down-regulation of translation, but enables specific translation of the transcription factor Gcn4 (*Hinnebusch, 2005*). Gcn4 then induces transcription of genes required for amino

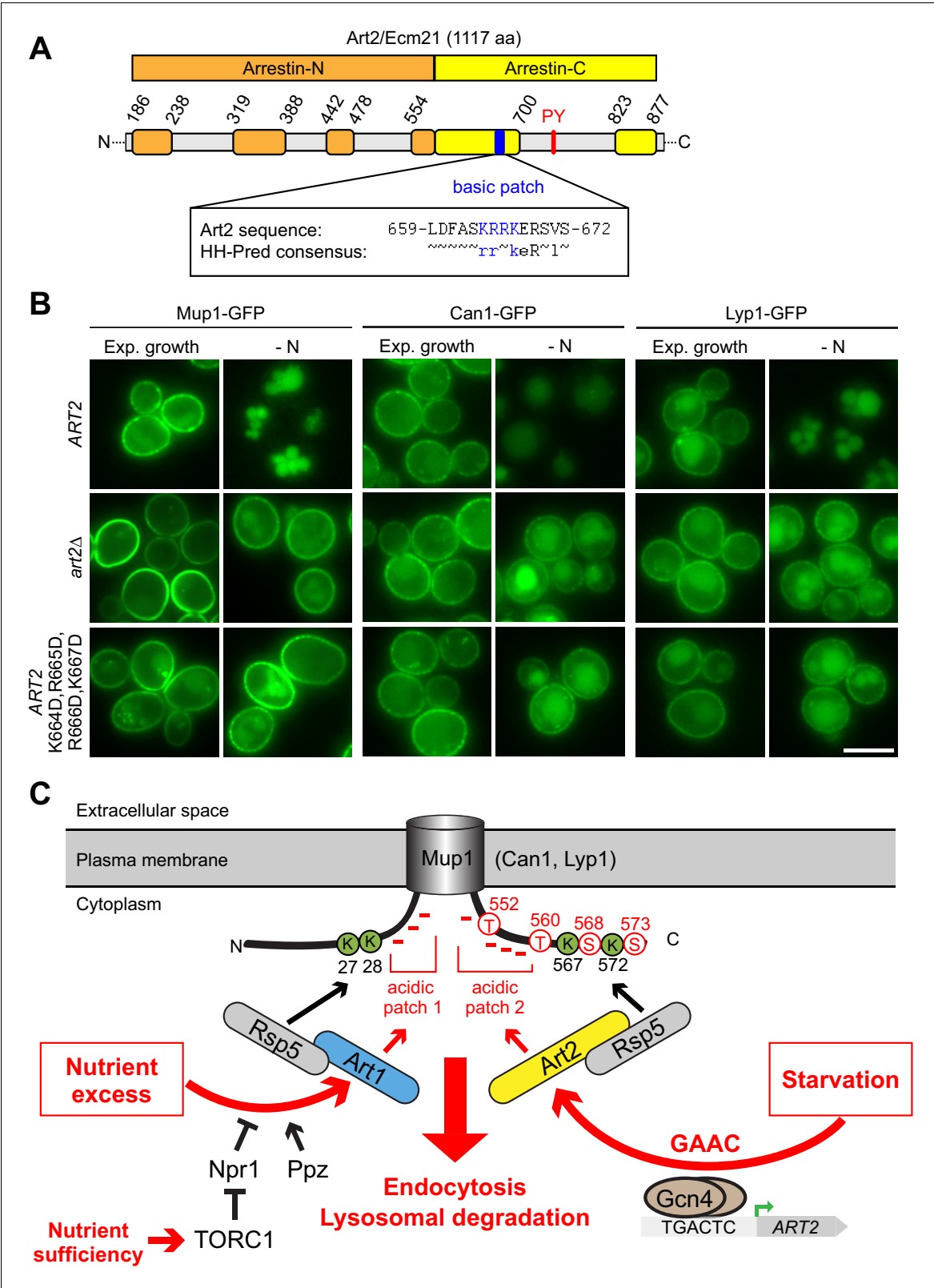

**Figure 7.** A basic patch of Art2 promotes the starvation-induced endocytosis of Mup1, Can1 and Lyp1. (**A**) Scheme of Art2 topology with arrestin-N domain in orange, arrestin-C domain in yellow and tails and interspersed extended loops (*Baile et al., 2019*) in light grey. The basic amino acid residues shown in blue mediate the starvation-induced endocytosis of Mup1, Can1 and Lyp1 (numbers correspond to amino acid positions in the Art2 sequence). Below is aligned the HHpred consensus sequence for Art2/Ecm21 derived from three HHblits iterations (*Zimmermann et al., 2018*),
*Figure 7 continued on next page*

Figure 7 continued

suggesting conservation of the basic patch in the Art2 protein family. (B) Live-cell fluorescence microscopy analysis of *art2Δ* cells expressing *MUP1-GFP*, *CAN1-GFP* or *LYP1-GFP* and pRS416-*ART2-WT*, empty vector or *pRS416-ART2 K664D,R665D,R666D,K667D*. Cells were starved (- N) for 6 hr after 24 hr exponential growth. (C) Scheme for the regulation of the substrate- and starvation-induced endocytosis of Mup1. During substrate excess TORC1 inhibits the Npr1 kinase which otherwise would phosphorylate and inhibit Art1. Art1-Rsp5 becomes dephosphorylated and subsequently binds the acidic patch 1 at the N-terminal tail of Mup1, leading to the ubiquitination of K27 and K28 and degradation of Mup1 (left). During amino acid and nitrogen starvation, the general amino acid control (GAAC) pathway upregulates the ubiquitin ligase adaptor Art2 via the transcriptional regulator Gcn4. The ensuing Art2-Rsp5 complex binds with its basic patch to the acidic patch 2 of Mup1 leading to the ubiquitination of K567 and K572 and degradation of Mup1 (right). At the same time, starvation causes TORC1 inhibition and activation of Npr1 and inhibits Art1-dependent ubiquitination. Ubiquitinated lysines (K) shown in green and phosphorylated serines (S) and threonines (T) in red with numbers corresponding to amino acid positions in the Mup1 sequence. Scale bars = 5 μm. See also *Figure 7—figure supplement 1*.

The online version of this article includes the following figure supplement(s) for figure 7:

**Figure supplement 1.** Additional characterization of the basic patch mutant of Art2.

acid biosynthesis and of *ART2*, which causes an increase in Art2 protein levels and thus formation of Art2-Rsp5 complexes. This appears as primary means to activate Art2, since unscheduled increase in Art2 protein levels was sufficient to drive Art2-dependent nutrient transporter endocytosis already in cells growing under rich conditions. When amino acids become available again, the system can efficiently reset. TORC1 is reactivated resulting in Art1 reactivation. Conversely, Gcn4 will become instable and rapidly degraded by the ubiquitin proteasome system (*Kornitzer et al., 1994*; *Meimoun et al., 2000*; *Irniger and Braus, 2003*), and thus, the transcription of *ART2* will cease. Interestingly, two de-ubiquitinating enzymes (Ubp2, Ubp15) de-ubiquitinate Art2 to influence its protein stability (*Ho et al., 2017*; *Kee et al., 2006*). Inhibiting their activity could provide additional control to repress Art2-dependent endocytosis in cells growing under rich conditions. Our screen identified also two de-ubiquinating enzymes, Doa4 and Ubp6 to be specifically required for starvation-induced endocytosis of Mup1. They could act directly on Art2 or Mup1 or help to maintain homeostasis of the ubiquitin pool during starvation.

Art2 is subject to extensive post-translational modification, including ubiquitination and phosphorylation. Database searches and our own proteomic experiments identified 68 phosphorylation sites and 20 ubiquitination sites in Art2 (data not shown) (*Swaney et al., 2013*; *Albuquerque et al., 2008*; *Holt et al., 2009*). How these modifications help to control the activity of Art2 remains a complex and open questions. Several arrestins were found to be phospho-inhibited in specific conditions (*MacGurn et al., 2011*; *Becuwe et al., 2012b*; *O'Donnell et al., 2013*; *Hovsepian et al., 2017*; *Merhi and André, 2012*; *Llopis-Torregrosa et al., 2016*), the common molecular basis of which is unknown. An exciting hypothesis would be that α-arrestin hyper-phosphorylation would add negative charges, and thereby prevent the recognition of acidic patches on transporters through electrostatic repulsion. Interestingly, our screen identified the pleitropic type 2A-related serine-threonine phosphatase Sit4 as a class one hit. Hence, Sit4 may be linked directly or indirectly to de-phosphorylation of Art2 and controlling its activity as reported recently for the Art2-dependent regulation of vitamin B1 transporters (*Savocco et al., 2019*).

Through the complementary activation of Art1 and Art2, cells can coordinate amino acid uptake through at least four high-affinity amino acid transporters with amino acid availability. The regulation of hexose transporters by glucose availability appears to be conceptually related, with distinct α−arrestin-Rsp5 complexes in charge of down-regulating the same transporters at various glucose concentrations with distinct mechanisms and kinetics (*Hovsepian et al., 2017*; *Nikko and Pelham, 2009*). In particular, the endocytosis of high-affinity hexose transporters during glucose starvation involves Art8, the closest paralogue of Art2, whose expression is also controlled by nutrient-regulated transcription (*Hovsepian et al., 2017*). Altogether, a picture emerges in which the transcriptional control of α−arrestin expression by nutrient-signaling pathways is critical to cope with nutrient depletion. This becomes part of the complex regulation of the ART-Rsp5 ubiquitin ligase network.

Our work also extends on previous findings regarding the determinants of α−arrestin/transporter interaction, indicating communalities between starvation- and substrate-induced endocytosis. Art1-Rsp5 and Art2-Rsp5 complexes both recognize specific acidic sequences on Mup1 (*Figure 7C*). Activated Art1-Rsp5 complexes recognize an acidic stretch in the N-terminus of Mup1 and Can1, close to the first transmembrane domain. The exposure of these N-terminal acidic patches is linked to

substrate transport and conformational transitions in the transporters from the outward-open to the inward-open conformation, which in Mup1 also includes the so-called 'C-plug' (aa 520–543) (*Busto et al., 2018*; *Guiney et al., 2016*; *Gournas et al., 2017*). This conformational switch drives lateral re-localization of Mup1 and Can1 into a disperse PM compartment, where they are ubiquitinated by Art1-Rsp5 (*Gournas et al., 2018*; *Busto et al., 2018*). Art2 recognizes specifically an acidic patch in the C-terminal tail of Mup1, and thereby directs Rsp5 to ubiquitinate two juxtaposed C-terminal lysine residues. The C-plug is very close to the C-terminal acidic patch but is not part of the C-terminal Mup1 degron. We speculate that in the absence of nutrients AATs will spend more time in the outward open state with the C-Plug in place. In this state, activated Art2-Rsp5 complexes can still engage the C-terminal acidic patches. Hence, toggling Art1/Art2 activation in combination with accessibility of N- or C-terminal acidic sorting signals in AATs, in part regulated by their conformational state, must fall together to allow selective endocytosis.

An additional layer of regulation for endocytosis is provided by phosphorylation of AATs close to the acidic sorting signal. At the moment, we can only speculate about the kinase responsible for the phosphorylation of the C-terminal serine or threonine sites of Mup1. Perhaps, constitutive PM-associated kinases such as the yeast casein kinase one pair (Yck1/2) are involved, which are known to recognize rather acidic target sequences and to regulate endocytosis (*Hicke et al., 1998*; *Paiva et al., 2009*; *Nikko et al., 2008*; *Marchal et al., 2002*).

α-Arrestins lack the polar core in the arrestin domain that is used for cargo interactions in β-arrestins (*Aubry et al., 2009*; *Polekhina et al., 2013*). Instead Art1-Rsp5 and Art2-Rsp5 complexes each use a basic region in their arrestin C-domain to detect the acidic sorting signal in their client AATs. Studies on the interaction between GPCRs and β-arrestins revealed a multimodal network of flexible interactions: The N-domain of β-arrestin interacts with phosphorylated regions of the GPCR, their finger loop inserts into the transmembrane domain bundle of the GPCR and loops at the C-terminal edge of β-arrestin engage the membrane (*Staus et al., 2020*; *Huang et al., 2020*). Perhaps, a similar concept also holds true for α-arrestins. This is not unlikely given that their arrestin fold appears to be interspersed with disordered loops and very long, probably unstructured N- and/or C-terminal tails, some of which participate in cargo recognition or membrane interactions (*Baile et al., 2019*).

Despite the possible plasticity in substrate interactions, the selectivity of Art1-Rsp5 and Art2-Rsp5 complexes in ubiquitinating lysine residues proximal to the acidic patches of Mup1 is remarkable. Mup1 has 19 lysine residues at the cytoplasmic side: four at the N-terminal tail, six in the C-terminal tail and nine in the intracellular loops of the pore domain. Yet, Art1-Rsp5 complexes only ubiquitinate K27 and K28, whereas Art2-Rsp5 complexes only ubiquitinate K567 and K572. Even in the Hxt3-Mup1-C chimeric protein, Art2-Rsp5 complexes ubiquitinated only the lysine residues close to the acidic patch, despite six further lysine residues in the directly adjacent C-terminal tail of Hxt3. How is this possible? We speculate that these two α-arrestin-Rsp5 complexes orient the HECT domain of Rsp5 with high precision toward the lysine residues that are spatially close to the acidic patches. Once ubiquitinated, the AAT can engage the endocytic machinery to be removed from the PM.

In conclusion, Art1-Rsp5 complexes act rapidly to prevent the accumulation of excess amino acids, whereas the Art2-Rsp5 complexes help to degrade idle high-affinity amino acid transporters over longer periods of starvation to recycle their amino acid content. Starvation-induced endocytosis and the subsequent degradation of membrane proteins is required to maintain intracellular amino acid homeostasis (*Müller et al., 2015*; *Jones et al., 2012*). As such, it is well suited that Art2 activity and thus starvation-induced endocytosis of AATs is co-regulated and coordinated with de novo amino acid biosynthesis via the GAAC pathway. The down-regulation of AATs together with glucose transporters and further PM proteins could also free up domains at the PM that are populated by selective nutrient transporters (*Spira et al., 2012*; *Grossmann et al., 2008*) for transporters with broader substrate specificity such as the general amino acid permease Gap1 and the ammonium transporter Mep2, which are strongly up-regulated during starvation. Hence starvation-induced endocytosis could prepare cells – anticipatory – for non-selective nutrient acquisition, as soon as nutrients become available again. Consistent with this notion and a role of Art2 during starvation, yeast cells with mutations in *ART2* show an altered adaptation to repetitive cycles of starvation (*Wloch-Salamon et al., 2017*). However, at the moment, it is not clear how these physiological defects are linked to the role of Art2 in starvation-induced endocytosis. Hence, despite the

emergence of regulatory mechanisms, we are still far from understanding how each individual ART-Rsp5 complex controls nutrient acquisition and how this process translates into fine-tuning of metabolism, cell growth and survival. Even more unclear is how the activities of the entire ART-Rsp5 network are harmonized to achieve a coordinated cellular response to nutrient availability. Since a more global disruption of the ART-Rsp5 network causes proteotoxic stress with strong growth defects (*Nikko and Pelham, 2009*; *Zhao et al., 2013*; *Hein et al., 1995*), addressing these important questions will require new approaches.

Altogether, our results provide a better understanding of the molecular mechanisms that couple metabolic signaling and nutrient availability to nutrient transporter endocytosis in budding yeast. This might provide basic insights and principles for the vastly more complex regulation of the nutrient transporter repertoire in human cells under normal as well as pathophysiological conditions.

# Materials and methods

## Key resources table

| Reagent type (species) or resource | Designation | Source or reference | Identifiers | Additional information |
|---|---|---|---|---|
| Strain, strainbackground *Saccharomyces cerevisiae* | SEY6210 | *Robinson et al., 1988* | SEY6210 | Parental yeast strain (genotype: *MATα leu2-3,112 ura3-52 his3-Δ200 trp1-Δ901 suc2-Δ9 lys2-801 GAL*), obtained from the Emr lab |
| Strain, strain background *Saccharomyces cerevisiae* | SEY6210.1 | *Robinson et al., 1988* | SEY6210.1 | Parental yeast strain (genotype: *MATa leu2-3, 112 ura3-52 his3-Δ200 trp1-Δ901 suc2-Δ9 lys2-801 GAL*), obtained from the Emr lab |
| Strain, strain background *Saccharomyces cerevisiae* | SEY6210 *rsp5Δ::HIS3 pRS415-6xHIS-TEV-3xFLAG-RSP5* | This paper | DTY557 | *RSP5* wild type yeast strain, obtained from the Emr lab (previously unpublished) |
| Strain, strain background *Saccharomyces cerevisiae* | SEY6210 *rsp5Δ::HIS3 pRS415-6xHIS-TEV-3xFLAG-RSP5-WW1\** | This paper | DTY558 | *RSP5* mutant yeast strain, obtained from the Emr lab (previously unpublished) |
| Strain, strain background *Saccharomyces cerevisiae* | SEY6210 *rsp5Δ::HIS3 pRS415-6xHIS-TEV-3xFLAG-RSP5-WW2\** | This paper | DTY559 | *RSP5* mutant yeast strain, obtained from the Emr lab (previously unpublished) |
| Strain, strain background *Saccharomyces cerevisiae* | SEY6210 *rsp5Δ::HIS3 pRS415-6xHIS-TEV-3xFLAG-RSP5-WW3\** | This paper | DTY560 | *RSP5* mutant yeast strain, obtained from the Emr lab (previously unpublished) |

*Continued on next page*

*Continued*

| Reagent type (species) or resource | Designation | Source or reference | Identifiers | Additional information |
|---|---|---|---|---|
| Strain, strain background *Saccharomyces cerevisiae* | SEY6210 *art4Δ:: HIS3 HXT3-GFP::TRP1* | This paper | ISY001 | Mutant yeast strain generated by homologous recombination (for PCR primer see *Supplementary file 4*) |
| Strain, strain background *Saccharomyces cerevisiae* | SEY6210 *art2Δ:: HIS3 HXT3-GFP::TRP1* | This paper | ISY013 | Mutant yeast strain generated by homologous recombination (for PCR primer see *Supplementary file 4*) |
| Strain, strain background *Saccharomyces cerevisiae* | SEY6210 *MUP1 D549,551,555R, E554R-GFP::HIS3* | This paper | ISY018 | Mutant yeast strain generated by homologous recombination (for PCR primer see *Supplementary file 4*) |
| Strain, strain background *Saccharomyces cerevisiae* | SEY6210 *HXT3-MUP1-C AA545-574 D549, 551,555R,E554R-GFP::TRP1* | This paper | ISY025 | Mutant yeast strain generated by homologous recombination (for PCR primer see *Supplementary file 4*) |
| Strain, strain background *Saccharomyces cerevisiae* | SEY6210.1 *art4Δ::HIS3 HXT3-MUP1-C AA545-574 D549, 551,555R,E554R-GFP::TRP1* | This paper | ISY026 | Mutant yeast strain generated by homologous recombination (for PCR primer see *Supplementary file 4*) |
| Strain, strain background *Saccharomyces cerevisiae* | SEY6210 *art2Δ::HIS3 art4Δ::HIS3 HXT3-MUP1-C AA545-574 D549, 551,555R,E554R-GFP::TRP1* | This paper | ISY027 | Mutant yeast strain generated by homologous recombination and mating/tetrad dissection (for PCR primer see *Supplementary file 4*) |
| Strain, strain background *Saccharomyces cerevisiae* | SEY6210 *art2Δ::HIS3 HXT3-MUP1-C AA545-574 D549, 551,555R,E554R-GFP::TRP1* | This paper | ISY028 | Mutant yeast strain generated by homologous recombination (for PCR primer see *Supplementary file 4*) |
| Strain, strain background *Saccharomyces cerevisiae* | SEY6210 *HXT3-MUP1-C AA545-574-GFP::TRP1* | This paper | ISY029 | Mutant yeast strain generated by homologous recombination (for PCR primer see *Supplementary file 4*) |
| Strain, strain background *Saccharomyces cerevisiae* | SEY6210.1 *art4Δ::HIS3 HXT3-MUP1-C AA545-574-GFP::TRP1* | This paper | ISY030 | Mutant yeast strain generated by homologous recombination (for PCR primer see *Supplementary file 4*) |

*Continued*

| Reagent type (species) or resource | Designation | Source or reference | Identifiers | Additional information |
|---|---|---|---|---|
| Strain, strain background *Saccharomyces cerevisiae* | SEY6210 *art2Δ::HIS3 art4Δ::HIS3 HXT3-MUP1-C AA545-574-GFP::TRP1* | This paper | ISY031 | Mutant yeast strain generated by homologous recombination and mating/tetrad dissection (for PCR primer see *Supplementary file 4*) |
| Strain, strain background *Saccharomyces cerevisiae* | SEY6210 *art2Δ::HIS3 HXT3-MUP1-C AA545-574-GFP::TRP1* | This paper | ISY032 | Mutant yeast strain generated by homologous recombination (for PCR primer see *Supplementary file 4*) |
| Strain, strain background *Saccharomyces cerevisiae* | SEY6210 *CAN1 D567, 575R,E569,574R-GFP::HIS3* | This paper | ISY033 | Mutant yeast strain generated by homologous recombination (for PCR primer see *Supplementary file 4*) |
| Strain, strain background *Saccharomyces cerevisiae* | SEY6210 *HXT3-MUP1-C AA545-574 K567,572R-GFP::TRP1* | This paper | ISY042 | Mutant yeast strain generated by homologous recombination (for PCR primer see *Supplementary file 4*) |
| Strain, strain background *Saccharomyces cerevisiae* | SEY6210.1 *art4Δ::HIS3 HXT3-MUP1-C AA545-574 K567,572R-GFP::TRP1* | This paper | ISY053 | Mutant yeast strain generated by homologous recombination (for PCR primer see *Supplementary file 4*) |
| Strain, strain background *Saccharomyces cerevisiae* | SEY6210 *art2Δ::HIS3 art4Δ::HIS3 HXT3-MUP1-C AA545-574 K567, 572R-GFP::TRP1* | This paper | ISY054 | Mutant yeast strain generated by homologous recombination and mating/tetrad dissection (for PCR primer see *Supplementary file 4*) |
| Strain, strain background *Saccharomyces cerevisiae* | SEY6210 *art2Δ::HIS3 HXT3-MUP1-C AA545-574 K567,572R-GFP::TRP1* | This paper | ISY055 | Mutant yeast strain generated by homologous recombination (for PCR primer see *Supplementary file 4*) |
| Strain, strain background *Saccharomyces cerevisiae* | SEY6210 *art2Δ::HIS3 art4::HIS3 HXT3-GFP::TRP1* | This paper | ISY061 | Mutant yeast strain generated by homologous recombination and mating/tetrad dissection (for PCR primer see *Supplementary file 4*) |
| Strain, strain background *Saccharomyces cerevisiae* | SEY6210.1 *gcn1Δ::HIS3* | This paper | JZY001 | Mutant yeast strain generated by homologous recombination (for PCR primer see *Supplementary file 4*) |

*Continued on next page*

*Continued*

| Reagent type (species) or resource | Designation | Source or reference | Identifiers | Additional information |
|---|---|---|---|---|
| Strain, strain background *Saccharomyces cerevisiae* | SEY6210.1 *gcn3Δ::HIS3* | This paper | JZY002 | Mutant yeast strain generated by homologous recombination (for PCR primer see *Supplementary file 4*) |
| Strain, strain background *Saccharomyces cerevisiae* | SEY6210.1 *gcn5Δ::HIS3* | This paper | JZY003 | Mutant yeast strain generated by homologous recombination (for PCR primer see *Supplementary file 4*) |
| Strain, strain background *Saccharomyces cerevisiae* | SEY6210.1 *gcn4Δ::HIS3* | This paper | OSY250 | Mutant yeast strain generated by homologous recombination (for PCR primer see *Supplementary file 4*) |
| Strain, strain background *Saccharomyces cerevisiae* | SEY6210.1 *gcn2Δ::HIS3* | This paper | OSY354 | Mutant yeast strain generated by homologous recombination (for PCR primer see *Supplementary file 4*) |
| Strain, strain background *Saccharomyces cerevisiae* | SEY6210.1 *art2Δ::HIS3 MUP1-GFP::TRP1* | This paper | TSY3 | Mutant yeast strain generated by homologous recombination (for PCR primer see *Supplementary file 4*) |
| Strain, strain background *Saccharomyces cerevisiae* | SEY6210 *art2Δ::HIS3* | This paper | TSY4 | Mutant yeast strain generated by homologous recombination (for PCR primer see *Supplementary file 4*) |
| Strain, strain background *Saccharomyces cerevisiae* | SEY6210.1 *art1Δ::HIS3* | This paper | VIY011 | Mutant yeast strain generated by homologous recombination (for PCR primer see *Supplementary file 4*) |
| Strain, strain background *Saccharomyces cerevisiae* | SEY6210.1 *art2Δ::TRP1* | This paper | VIY012 | Mutant yeast strain generated by homologous recombination (for PCR primer see *Supplementary file 4*) |
| Strain, strain background *Saccharomyces cerevisiae* | SEY6210 *ART2-6xHIS-TEV-3xFLAG::HIS3* | This paper | VIY030 | Mutant yeast strain generated by homologous recombination (for PCR primer see *Supplementary file 4*) |
| Strain, strain background *Saccharomyces cerevisiae* | SEY6210 *ART1-6xHIS-TEV-3xFLAG::HIS3* | This paper | VIY031 | Mutant yeast strain generated by homologous recombination (for PCR primer see *Supplementary file 4*) |

*Continued on next page*

*Continued*

| Reagent type (species) or resource | Designation | Source or reference | Identifiers | Additional information |
|---|---|---|---|---|
| Strain, strain background *Saccharomyces cerevisiae* | SEY6210 *CAN1-GFP::HIS3* | This paper | VIY036 | Mutant yeast strain generated by homologous recombination (for PCR primer see *Supplementary file 4*) |
| Strain, strain background *Saccharomyces cerevisiae* | SEY6210 *HXT3-GFP::TRP1* | This paper | VIY042 | Mutant yeast strain generated by homologous recombination (for PCR primer see *Supplementary file 4*) |
| Strain, strain background *Saccharomyces cerevisiae* | SEY6210 *MUP1-GFP::TRP1* | This paper | VIY054 | Mutant yeast strain generated by homologous recombination (for PCR primer see *Supplementary file 4*) |
| Strain, strain background *Saccharomyces cerevisiae* | SEY6210.1 *art2Δ::TRP1 INA1-eGFP::HIS3* | This paper | OSY924 | Mutant yeast strain generated by homologous recombination (for PCR primer see *Supplementary file 4*) |
| Strain, strain background *Saccharomyces cerevisiae* | SEY6210 *MUP1 S568, 573A-GFP::TRP1* | This paper | VIY059 | Mutant yeast strain generated by homologous recombination (for PCR primer see *Supplementary file 4*) |
| Strain, strain background *Saccharomyces cerevisiae* | SEY6210.1 *art2Δ::TRP1 CAN1-GFP::HIS3* | This paper | VIY062 | Mutant yeast strain generated by homologous recombination (for PCR primer see *Supplementary file 4*) |
| Strain, strain background *Saccharomyces cerevisiae* | SEY6210.1 *art2Δ::TRP1 HXT2-GFP::HIS3* | This paper | VIY063 | Mutant yeast strain generated by homologous recombination (for PCR primer see *Supplementary file 4*) |
| Strain, strain background *Saccharomyces cerevisiae* | SEY6210.1 *art2Δ::TRP1 TAT2-GFP::HIS3* | This paper | VIY064 | Mutant yeast strain generated by homologous recombination (for PCR primer see *Supplementary file 4*) |
| Strain, strain background *Saccharomyces cerevisiae* | SEY6210.1 *art2Δ::TRP1 HXT1-GFP::HIS3* | This paper | VIY065 | Mutant yeast strain generated by homologous recombination (for PCR primer see *Supplementary file 4*) |
| Strain, strain background *Saccharomyces cerevisiae* | SEY6210.1 *art2Δ::TRP1 LYP1-GFP::HIS3* | This paper | VIY068 | Mutant yeast strain generated by homologous recombination (for PCR primer see *supplementary file 4*) |

*Continued on next page*

*Continued*

| Reagent type (species) or resource | Designation | Source or reference | Identifiers | Additional information |
|---|---|---|---|---|
| Strain, strain background *Saccharomyces cerevisiae* | SEY6210 *MUP1 K567,572R-GFP::TRP1* | This paper | VIY073 | Mutant yeast strain generated by homologous recombination (for PCR primer see *Supplementary file 4*) |
| Strain, strain background *Saccharomyces cerevisiae* | SEY6210 *MUP1 T552,560A-GFP::TRP1* | This paper | VIY076 | Mutant yeast strain generated by homologous recombination (for PCR primer see *Supplementary file 4*) |
| Strain, strain background *Saccharomyces cerevisiae* | SEY6210.1 *art1Δ::HIS3 MUP1-GFP::TRP1* | This paper | VIY079 | Mutant yeast strain generated by homologous recombination (for PCR primer see *Supplementary file 4*) |
| Strain, strain background *Saccharomyces cerevisiae* | SEY6210 *MUP1 K27, 28R-GFP::TRP1* | This paper | VIY082 | Mutant yeast strain generated by homologous recombination (for PCR primer see *Supplementary file 4*) |
| Strain, strain background *Saccharomyces cerevisiae* | SEY6210.1 *art2Δ::TRP1 MUP1-GFP::HIS3* | This paper | VIY092 | Mutant yeast strain generated by homologous recombination (for PCR primer see *Supplementary file 4*) |
| Strain, strain background *Saccharomyces cerevisiae* | SEY6210 *MUP1 K27, 28,567,572R-GFP::TRP1* | This paper | VIY093 | Mutant yeast strain generated by homologous recombination (for PCR primer see *Supplementary file 4*) |
| Strain, strain background *Saccharomyces cerevisiae* | SEY6210.1 *art1Δ::HIS3 art2Δ::TRP1* | This paper | VIY107 | Mutant yeast strain generated by homologous recombination (for PCR primer see *Supplementary file 4*) |
| Strain, strain background *Saccharomyces cerevisiae* | SEY6210 *MUP1-HA-TEV-GFP K1R::TRP1* | This paper | VIY200 | Mutant yeast strain generated by homologous recombination (for PCR primer see *Supplementary file 4*) |
| Strain, strain background *Saccharomyces cerevisiae* | SEY6210 *MUP1 S568,573A-HA-TEV-GFP K1R::TRP1* | This paper | VIY242 | Mutant yeast strain generated by homologous recombination (for PCR primer see *Supplementary file 4*) |
| Strain, strain background *Saccharomyces cerevisiae* | SEY6210 *MUP1 T552,560A-HA-TEV-GFP K1R::TRP1* | This paper | VIY255 | Mutant yeast strain generated by homologous recombination (for PCR primer see *Supplementary file 4*) |

Continued

| Reagent type (species) or resource | Designation | Source or reference | Identifiers | Additional information |
|---|---|---|---|---|
| Strain, strain background *Saccharomyces cerevisiae* | SEY6210 *MUP1 G78N-HA-TEV-GFP K1R::TRP1* | This paper | VIY263 | Mutant yeast strain generated by homologous recombination (for PCR primer see *supplementary file 4*) |
| Strain, strain background *Saccharomyces cerevisiae* | SEY6210.1 *ART2-6xHIS-TEV-3xFLAG::HIS3 gcn4Δ::HIS3* | This paper | VIY291 | Mutant yeast strain generated by homologous recombination and mating/tetrad dissection (for PCR primer see *supplementary file 4*) |
| Strain, strain background *Saccharomyces cerevisiae* | SEY6210 *MUP1 D43A, G47A,Q49A, T52A,L54A-HA-TEV-GFP K1R::TRP1* | This paper | VIY295 | Mutant yeast strain generated by homologous recombination (for PCR primer see *supplementary file 4*) |
| Strain, strain background *Saccharomyces cerevisiae* | SEY6210 *MUP1 T6,25,26, 34A,S9,22, 23,24,31, 33,42A,K16, 27,28R-HA-TEV-GFP K1R::TRP1* | This paper | VIY335 | Mutant yeast strain generated by homologous recombination (for PCR primer see *supplementary file 4*) |
| Strain, strain background *Saccharomyces cerevisiae* | SEY6210 *MUP1-GFP::HIS3* | This paper | YSM24 | Mutant yeast strain generated by homologous recombination (for PCR primer see *supplementary file 4*) |
| Recombinant DNA reagent | pRS413 (Plasmid) | *Sikorski and Hieter, 1989* | pRS413 | Empty centromer vector *HIS3* |
| Recombinant DNA reagent | pRS414 (Plasmid) | *Sikorski and Hieter, 1989* | pRS414 | Empty centromer vector *TRP1* |
| Recombinant DNA reagent | pRS415 (Plasmid) | *Sikorski and Hieter, 1989* | pRS415 | Empty centromer vector *LEU2* |
| Recombinant DNA reagent | pRS416 (Plasmid) | *Sikorski and Hieter, 1989* | pRS416 | Empty centromer vector *URA3* |
| Recombinant DNA reagent | YCp5O-*GCN4* (Plasmid) | *Hinnebusch, 1985* | p180 | Wild type *GCN4* (centromeric plasmid, *URA3* selection) |
| Recombinant DNA reagent | YCp5O-*GCN4*[C] (Plasmid) | *Mueller et al., 1987* | p227 | Constitutively active *GCN4* (centromeric plasmid, *URA3* selection) |
| Recombinant DNA reagent | pRS416-*CAN1-GFP* (Plasmid) | *Lin et al., 2008* | pCHL571 | GFP-tagged version of *CAN1* (centromeric plasmid, *URA3* selection) |
| Recombinant DNA reagent | pRS416-*MUP1-GFP* (Plasmid) | *Lin et al., 2008* | pCHL642 | GFP-tagged version of *MUP1* (centromeric plasmid, *URA3* selection) |

*Continued*

| Reagent type (species) or resource | Designation | Source or reference | Identifiers | Additional information |
|---|---|---|---|---|
| Recombinant DNA reagent | pRS416-*ART2-6xHIS-TEV-3xFLAG* (Plasmid) | This paper | pIS001 | HTF-tagged version of *ART2* (centromeric plasmid, *URA3* selection; for cloning primer see *supplementary file 4*) |
| Recombinant DNA reagent | pRS416-*ART2 P748, 749A,Y750A-6xHIS-TEV-3xFLAG* (Plasmid) | This paper | pIS003 | PxY mutant of *ART2-HTF* (centromeric plasmid, *URA3* selection; for cloning primer see *Supplementary file 4*) |
| Recombinant DNA reagent | pRS416-*ART2 K664, 667D,R665, 666D-6xHIS-TEV-3xFLAG* (Plasmid) | This paper | pIS004 | Basic patch mutant of *ART2-HTF* (centromeric plasmid, *URA3* selection; for cloning primer see *supplementary file 4*) |
| Recombinant DNA reagent | pRS415-*ART2-6xHIS-TEV-3xFLAG* (Plasmid) | This paper | pJZ001 | HTF-tagged version of *ART2* (centromeric plasmid, *LEU2* selection; for cloning primer see *supplementary file 4*) |
| Recombinant DNA reagent | pRS415-*pART2\*-ART2-6xHIS-TEV-3xFLAG* (Plasmid) | This paper | pJZ002 | Promotor mutant of *ART2-HTF* (centromeric plasmid, *LEU2* selection; for cloning primer see *supplementary file 4*) |
| Recombinant DNA reagent | pRS416-*pART2-ART1-6xHIS-TEV-3xFLAG* (Plasmid) | This paper | pOS258 | *ART1-HTF* expressed under *ART2* promoter (centromeric plasmid, *URA3* selection; for cloning primer see *supplementary file 4*) |
| Recombinant DNA reagent | pRS416-*MUP1-pHluorin* (Plasmid) | This paper | pLZ78 | pHluorin-tagged version of *MUP1*, obtained from the Emr lab (previously unpublished) (centromeric plasmid, *URA3* selection) |
| Recombinant DNA reagent | pRS415-*MUP1-GFP* (Plasmid) | *Schmidt et al., 2017* | pMM02 | GFP-tagged version of *MUP1* (centromeric plasmid, *LEU2* selection) |

*Continued on next page*

*Continued*

| Reagent type (species) or resource | Designation | Source or reference | Identifiers | Additional information |
|---|---|---|---|---|
| Recombinant DNA reagent | pRS416-*FUR4-GFP* (Plasmid) | *MacGurn et al., 2011* | pSR21 | GFP-tagged version of *FUR4* (centromeric plasmid, *URA3* selection) |
| Recombinant DNA reagent | pRS415-*ART1* (Plasmid) | This paper | pVI009 | untagged version of *ART1* (centromeric plasmid, *LEU2* selection; for cloning primer see *Supplementary file 4*) |
| Recombinant DNA reagent | pRS416-*pTDH3-ART2-6xHIS-TEV-3xFLAG* (Plasmid) | This paper | pVI013 | Overexpression of *ART2-HTF* (centromeric plasmid, *URA3* selection; for cloning primer see *Supplementary file 4*) |
| Antibody | α-FLAG M2 (Mouse monoclonal) | Sigma, Austria | F1804 (RRID:AB_259529) | WB (1:10.000) |
| Antibody | α-GFP IgG1K (Mouse monoclonal) | Sigma, Austria | 11814460001 (RRID:AB_390913) | WB (1:1000) |
| Antibody | α-Pgk1 22C5D8 (Mouse monoclonal) | Invitrogen, USA | 459250 (RRID:AB_2532235) | WB (1:10.000) |
| Antibody | α-ubiquitin P4D1 (Mouse monoclonal) | Santa Cruz Biotechnology, USA | 3936S (RRID:AB_628423) | WB (1:500) |
| Antibody | goat a-mouse (polyclonal)-peroxidase | Sigma | A4416 (RRID:AB_258167) | WB (1:5000) |
| Commercial assay or kit | RevertAid First Strand cDNA Synthesis Kit | Thermo Fisher, USA | K1622 | |
| Commercial assay or kit | RNeasy Mini Kit | Qiagen, Germany | 74104 | |
| Commercial assay or kit | TaqMan Assay ECM21/ART2 | Thermo Fisher, USA | Sc04099967_s1 | |
| Commercial assay or kit | TaqMan Assay PGK1 | Thermo Fisher, USA | Sc04104844_s1 | |
| Commercial assay or kit | TaqMan Gene Expression Master Mix | Thermo Fisher, USA | 4369016 | |
| Commercial assay or kit | Non-essential gene deletion strain collection in BY4742 | Open Biosystems, USA | YSC1054 | |
| Chemical compound, drug | GFP-Trap_MA Magnetic Agarose beads | ChromoTek, Germany | gtma-20 | |

*Continued on next page*

*Continued*

| Reagent type (species) or resource | Designation | Source or reference | Identifiers | Additional information |
|---|---|---|---|---|
| Chemical compound, drug | Nonidet P 40 | Fluka, Germany | 74385 | |
| Chemical compound, drug | Yeast Nitrogen Base without amino acids | Roth, Germany | HP26.1 | |
| Chemical compound, drug | Yeast Nitrogen Base without amino acids and ammonium sulfate | VWR, USA | J630-100G | |
| Software, algorithm | Fiji | *Schindelin et al., 2012* | Version 1.0 | |
| Software, algorithm | GuavaSoft | Luminex | Version 2.7 | |
| Software, algorithm | Illustrator CS5.1 | Adobe | Version 15.1.0 (RRID:SCR_010279) | |
| Software, algorithm | ImageJ2 | *Rueden et al., 2017* | Version 2.0.0-rc49/ 1.51 hr (RRID:SCR_003070) | |
| Software, algorithm | Photoshop CS5 | Adobe | Version 12.0.4 × 64 (RRID:SCR_014199) | |
| Software, algorithm | PikoReal | Thermo Fisher | version 2.2 | |
| Software, algorithm | VisiView | Visitron | version 2.1.4 | |

## Yeast strains, media and growth conditions

Yeast strains used for the microscopy screen for starvation-responsive endocytosis cargoes were mainly derived from the Yeast C-terminal GFP Collection (*Huh et al., 2003*) with addition of further C-terminally-tagged transporters in BY4741 (*MATa his3Δ1 leu2Δ0 met15Δ0 ura3Δ0*) and SEY6210 (*MATα leu2-3,112 ura3-52 his3-200 trp1-901 lys2-801 suc2-9*) strain background (*Supplementary file 1*). The flow cytometry screen for genes affecting the starvation-induced endocytosis of Mup1-pHluorin was performed using the non-essential gene deletion strain collection purchased from Open Biosystems (BY4742: *MATα his3Δ1 leu2Δ0 lys2Δ0 ura3Δ0*) transformed with pRS416 expressing Mup1-pHluorin. For all other experiments, the SEY6210 parental strain was used (see key resources table).

In all experiments, except the genome-wide flow cytometry screen, cells were cultivated in rich medium containing: 6,7 g/L Yeast Nitrogen Base without amino acids (#HP26.1, Roth, Germany), 20 mg/L adenine hemisulfate (#A3159-25G, Sigma, Austria), 20 mg/L arginine (#1655.1, Roth, Germany), 230 mg/L lysine monohydrate (#4207.1, Roth, Germany), 300 mg/L threonine (#T206.2, Roth, Germany), 30 mg/L tyrosine (#T207.1, Roth, Germany), 2% glucose (#X997.5, Roth, Germany). The rich medium was supplemented when required for the auxotrophic strains with: 20 mg/L histidine (#3852.1, Roth, Germany), 60 mg/L leucine (#3984.1, Roth, Germany), 20 mg/L tryptophan (#4858.2, Roth, Germany), 20 mg/L uracil (#7288.2, Roth, Germany), 20 mg/L methionine (#9359.1, Roth, Germany). Starvation medium contained 1,7 g/L Yeast Nitrogen Base without amino acids and ammonium sulfate (#J630-100G, VWR, USA) and 2% glucose (#X997.5, Roth, Germany).

Yeast cells were cultivated in rich medium and kept for at least 24 hr in exponential growth phase by dilution into fresh medium before onset of all experiments at mid-log phase. All cultivations were done at 26°C. Substrate-induced endocytosis of Mup1 was triggered by the addition of 20 mg/L methionine (#9359.1, Roth, Germany) and, unless otherwise stated, analyzed 1.5 hr after the

treatment. For starvation experiments, exponentially growing cells (0.4–0.6 OD$_{600}$/ml) were washed twice with and resuspended in starvation medium (1 OD$_{600nm}$/ml) and incubated for the indicated times.

## Genetic modifications and cloning

Genetic modifications were performed by PCR and/or homologous recombination using standard techniques. All chromosomal tags were introduced at the C-terminus of the target ORFs to preserve the endogenous five prime regulatory sequences. Chromosomally modified yeast strains were analyzed by genotyping PCR and/or DNA-sequencing. Plasmid-expressed genes including their endogenous or heterologous promoters were amplified from yeast genomic DNA and cloned into centromeric vectors (pRS series). All plasmids were analyzed by DNA-sequencing. Standard techniques were used for yeast transformation, mating and tetrad analysis or haploid selection. Yeast stains and plasmids are listed in the key resources table and primer in *Supplementary file 4*.

## Flow cytometry screen

The yeast non-essential knock out collection (YSC1054; Open Biosystems) was transformed in 96-well format using a standard lithium acetate/PEG-4000 yeast transformation protocol with a pRS416 plasmid encoding Mup1-pHluorin. Transformants were selected on agar plates, inoculated into 160 µl of rich medium (YNB medium lacking uracil as specified above containing 1.5x of all amino acids) and incubated for 14 hr shaking (180 rpm) at 26° in 96-well plates (#83.3924, Sarstedt, Germany). Each plate also contained the WT BY4742 parental strain and *art2Δ* as negative and positive controls. 60 µl of the overnight cultures were transferred to 96-deep-well plates (#951033405, 96/2000 µl, white border, Eppendorf, Germany) containing 600 µl rich medium and further incubated for 5 hr. At this point, 120 µl of culture referred to as 'exponentially growing' were transferred to standard 96-well plates and analyzed by flow cytometry. The remaining culture was harvested by centrifugation (1807 g; 3 min), the medium was aspirated, cells were washed twice with 600 µl starvation medium and recovered by centrifugation. Subsequently, cells were resuspended in 600 µl starvation medium and incubated shaking (180 rpm) for 18–22 hr. 200 µl nitrogen-starved culture were transferred to standard 96-well plates and analyzed by flow cytometry. All pipetting steps were performed using the MEA Robotic System (PhyNexus, USA). Flow cytometry was performed in 96-well format using the Guava easyCyte 8HT-System (Sr.No. 6735128143), EMD Millipore, Merck, Germany) with the following settings: Energy GRN: 72 plus (YEL = 8; RED = 8; NIR = 8; RED2 = 8; NIR2 = 8), green channel, forward scatter FSC = 14; sideward scatter SSC = 28; 15.000–20.000 counts/sample, acquire: 40 s, 3 s mix prior to acquisition. GuavaSoft 2.7 software was used for data analysis. The positive/negative cut-off was set for each plate empirically at the intercept of the log/starvation histograms of the WT and *art2Δ* controls (*art2Δ*, which emerged as a well-reproducible hit early in the screen, was included as a negative control in all further plates). All potential hits were re-examined by fluorescence microscopy. To this end, at least 100 starved cells were analyzed by fluorescence microscopy after starvation and the percentage of cells showing a degradation-deficient phenotype (Mup1-pHluorin at the plasma membrane, in small cytosolic objects, class E-like objects or small objects within vacuoles) of the total number of cells counted was calculated (*Supplementary file 2*). Strains with more than 45% cells with retained fluorescence after at least 18 hr of starvation were considered as hits. For a stringent final selection, we compared those hits to the original flow cytometry screen and finally only considered those in which at least once more the 30% Mup1-pHluorin fluorescence was also retained after starvation in the flow cytometry screen. In addition, most hits were also scored for methionine-induced endocytosis of Mup1-pHluorin. Hits were considered starvation-specific if the fluorescence was quenched in more than 67% of cells after 90 min of methionine treatment (20 µg/ml).

## Gene ontology analysis

Gene Ontology (GO) enrichment analysis (*Harris et al., 2004*) was performed using 128 genes listed in *Supplementary file 2*. They were mapped against the generic GO-Slim: process, generic GO-Slim: component and macromolecular complex terms: component GO sets using the GoSLIM mapper of the *Saccharomyces* genome database. We calculated the ratio of the observed (cluster frequency) vs. the expected number of genes (genome frequency) associated with the GO term,

referred to as enrichment over genome (*McClellan et al., 2007*; *Schmidt et al., 2019*). Only GO terms with more than one associated gene were reported. The full analysis is presented in *Supplementary file 3*.

## Fluorescence live cell wide-field microscopy

For microscopy, cells were concentrated by centrifugation and directly mounted onto glass slides. Live cell wide-field fluorescence microscopy was carried out using a Zeiss Axio Imager M1 equipped with a sola light engine LED light source (Lumencore), a 100x oil immersion objective (NA 1.45) standard GFP and mCherry fluorescent filters, a SPOT Xplorer CCD camera, and Visitron VisiView software (version 2.1.4). The brightness and contrast of the images in the figures were linearly adjusted using Photoshop CS5 (Adobe Version 12.0.4 $\times$ 64).

## Preparation of yeast whole cell protein extracts

10 $OD_{600nm}$ yeast cells were pelleted by centrifugation, resuspended in ice-cold water with 10% trichloroacetic acid (#T0699, Sigma, Austria), incubated on ice for at least 30 min and washed twice with ice cold acetone (#5025.5, Roth, Germany). The precipitate was resuspended in 200 µl extraction buffer (150 mM Tris-HCl, pH 6,8 (#443866G, VWR, USA); 6% SDS (#CN30.3, Roth, Germany), 6M urea (#51456–500G, Sigma, Austria), 6% glycerol (#3783.2, Roth, Germany), 3% β-mercaptoethanol (#M6250, Sigma, Austria), 0,01% bromophenol blue (#44305, BDH Laboratory Supplies, England)) and solubilized with 0.75–1 mm glass beads (#A554.1, Roth Germany) for 15 min at RT and subsequent heating at 42℃ for 30 min and 60℃ for 10 min.

## Western blot and immunodetection

Protein extracts from total cells and eluates from immuno-precipitations were separated by standard SDS-PAGE and transferred to PVDF membranes (#10600023, VWR, USA) for the detection of Mup1-GFP, Art2-HTF and Pgk1 or to nitrocellulose membranes (#10600004, VWR, USA) for the detection of ubiquitinated Mup1. PVDF membranes were stained with Coomassie Brilliant Blue R250 (#3862.2, Roth, Germany) for assessment of transfer and loading. Antibodies used in this study include: α-FLAG M2 (#F1804, Sigma, Austria), α-GFP IgG1K (#11814460001, Sigma, Austria), α-Pgk1 22C5D8 (#459250, Invitrogen, USA), α-ubiquitin P4D1 (#3936S, Santa Cruz Biotechnology, USA). Secondary antibody was goat α-mouse IgG peroxidase (#A4416, Sigma, Austria). Western blots were developed with Advansta Western bright ECL substrate (#541005 (K-12045-D50), Biozyme, Austria).

## Western blot quantification

WB signals were quantified by densitometry using ImageJ2 (Version 2.0.0-rc49/1.51 hr; RRID:SCR_003070) (*Rueden et al., 2017*), quantifications were exported to Microsoft Excel (Version 16.16.2; RRID:SCR_016137), normalized to the respective Pgk1 loading controls, and presented as mean ± standard deviation from at least three independent experiments. WT in exponential growth phase was set to 1. For the quantification of GFP-tag processing, full-length and clipped GFP WB signals were quantified from the same exposure and added to calculate the total GFP signal for each lane. The signals for full-length and clipped GFP were then reported as the percentage from the total GFP signal.

## Immunoprecipitation

Mup1-GFP immunoprecipitation protocol was adapted from *Hovsepian et al., 2017*. 50 $OD_{600nm}$ yeast cells were harvested by centrifugation and mechanically disrupted by glass bead lysis (0.75–1 mm) at 4℃ in 500 µl ice-cold RIPA lysis buffer (50 mM Tris-HCl, pH 7.5, 150 mM NaCl (#3957.5, Roth, Germany), 0.1% SDS, 2 mM EDTA (#ED-1KG, Sigma, Austria), 50 mM NaF (#SO0323, Scharlau, Spain), 1% Nonidet P 40 (#74385, Fluka, Germany), 0.5% Na-deoxycholate (#D6750-25G, Sigma, Austria), 1% glycerol) containing protease inhibitors (cOmplete Protease Inhibitor Cocktail (#11697498001, Sigma, Austria), yeast protease inhibitor cocktail (yPIC, #P8215-5ML, Sigma, Austria), 2 mM phenylmethylsulfonyl fluoride (PMSF, #P7626-5G, Sigma, Austria)) and 20 mM *N*-ethylmaleimide (NEM, #E3876-5G, Sigma, Austria) using four cycles of lysis (2 min), each separated by 2 min chilling on ice. The lysate was then rotated for 30 min at 4℃ for solubilization. 500 µl wash buffer (50 mM Tris-HCl, pH 7.5, 150 mM NaCl, 1% Nonidet P 40, 5% glycerol) containing 20 mM NEM

were added and the sample was mixed and further rotated for 1 hr at 4°C. After vortexing for 3 min, the lysate was centrifuged at 10000 g for 10 min at 4°C. The cleared lysate was then added to 25 µl of GFP-Trap_MA magnetic agarose beads (#gtma-20, ChromoTek, Germany) prewashed twice in 1 ml wash buffer. The sample was rotated for 5 hr at 4°C. The beads were collected using a magnetic rack and washed twice by rotating for 15 min at 4°C with 1.5 ml ice-cold wash buffer. The beads were further washed for 30 min at RT with 1.5 ml saline buffer (50 mM Tris-HCl, pH 7.5, 150 mM NaCl), containing 0.1% Tween-20 (#9127.1, Roth, Germany), resuspended in 50 µl 2x urea sample buffer (150 mM Tris-HCl, pH 6.8, 6M urea, 6% SDS, 0.01% bromophenol blue) and incubated for 30 min at 1400 rpm and 37°C in a thermomixer. Then, 50 µl boiling buffer (50 mM Tris-HCl, pH 7.5, 1 mM EDTA, 1%, 20% glycerol) was added and the sample was further incubated for 30 min at 1400 rpm and 42°C in a thermomixer. The resulting eluates were subjected to WB analysis. For mass spectrometry analysis of Mup1-GFP 300 $OD_{600}$ equivalents were used with the following modifications: Cells were lysed by with glass beads (3 × 1 min) at 4°C in 3 ml RIPA buffer (50 mM Tris/HCl, pH 7,5, 200 mM NaCl, 10% glycerol, 1% NP-40, 1% Na-Deoxycholate, 0,1% SDS, 2 mM EDTA, 0,1% Tween-20, 25 mM NaF, 1x PhosStop (#PHOSS-RO, Sigma, Germany), 1x Complete EDTA-free protease inhibitors, 0.67 mM DTT, 1x yPIC, 2 mM PMSF, 20 mM NEM. Subsequently another 3 ml of RIPA buffer were added and the lysate was sonicated 5 × 1 min at 4°C in a water bath sonicator. Lysates were incubated 30 min on ice and then centrifuged at 10,000 g for 10 min to remove debris. 80 ul of equilibrated GFP-trap beads were added to the supernatant and incubated rolling for 16 hr at 4°C. Beads were washed at 4°C 3 × 15 min with RIPA buffer supplemented with 500 mM NaCl and then 3 × 15 min with urea wash buffer (50 mM Tris/HCl pH 7.5, 100 mM NaCl, 4M Urea). Mup1-GFP was eluted from beads using urea sample buffer as described above. Eluates were separated via SDS-PAGE and Mup1-GFP was visualized by Coomassie staining. Slices were cut from the gel (including the visible Mup1-GFP protein bands and the region above containing ubiquitinated Mup1-GFP) and subjected for further mass spectrometry sample preparation.

## Mass spectrometry sample preparation and analysis

Coomassie-stained gel bands were excised from SDS-PAGE gels, reduced with dithiothreitol, alkylated with iodoacetamide and digested with trypsin (Promega) as previously described (*Faserl et al., 2019*). Tryptic digest were analyzed using an UltiMate 3000 RSCLnano-HPLC system coupled to a Q Exactive HF mass spectrometer (both Thermo Scientific, Bremen, Germany) equipped with a Nanospray Flex ionization source. The peptides were separated on a homemade fritless fused-silica micro-capillary column (75 µm i.d. x 280 µm o.d. x 10 cm length) packed with 3.0 µm reversed-phase C18 material. Solvents for HPLC were 0.1% formic acid (solvent A) and 0.1% formic acid in 85% acetonitrile (solvent B). The gradient profile was as follows: 0–4 min, 4% B; 4–57 min, 4–35% B; 57–62 min, 35–100% B, and 62–67 min, 100% B. The flow rate was 250 nL/min.

The Q Exactive HF mass spectrometer was operating in the data dependent mode selecting the top 20 most abundant isotope patterns with charge >1 from the survey scan with an isolation window of 1.6 mass-to-charge ratio (m/z). Survey full scan MS spectra were acquired from 300 to 1750 m/z at a resolution of 60,000 with a maximum injection time (IT) of 120 ms, and automatic gain control (AGC) target 1e6. The selected isotope patterns were fragmented by higher-energy collisional dissociation with normalized collision energy of 28 at a resolution of 30,000 with a maximum IT of 120 ms, and AGC target 5e5.

Data analysis was performed using Proteome Discoverer 4.1 (Thermo Scientific) with search engine Sequest. The raw files were searched against yeast database (orf_trans_all) with sequence of Mup1-GFP added. Precursor and fragment mass tolerance was set to 10 ppm and 0.02 Da, respectively, and up to two missed cleavages were allowed. Carbamidomethylation of cysteine was set as static modification. Oxidation of methionine, ubiquitination of lysine, and phosphorylation of serine threonine, and tyrosine were set as variable modifications. Peptide identifications were filtered at 1% false discovery rate.

## RNA isolation and quantitative PCR (RT-qPCR)

Exponentially growing or starved cells (40 $OD_{600nm}$) were harvested by centrifugation and immediately frozen in lq. $N_2$. Cell pellets were lysed with 1 mm glass beads in a FastPrep-24 homogenizer (MP biosciences) in Qiagen RLT buffer, and RNA was extracted using the RNeasy Mini Kit (#74104,

Qiagen, Germany). Yield and purity were determined photometrically. cDNA was prepared from 5 µg DNAse I-treated RNA using the RevertAid First Strand cDNA Synthesis Kit (#K1622, Thermo Fisher, USA) with oligo-dT primer according to the standard protocol. qPCR was performed in 10 µl scale with 4 µl of cDNA, 5 µl TaqMan Gene Expression Master Mix (#4369016, Thermo Fisher, USA) and 0.5 µl TaqMan probe on a PikoReal 96 Real-Time PCR System (Thermo Fisher, USA) with 7 min initial denaturation (95°C) and 40 cycles of 5 s 95°C and 30 s 60°C. TaqMan gene expression assays were from Thermo Fisher (*PGK1*: Sc04104844_s1; *ECM21/ART2*: Sc04099967_s1). All probes and primer anneal within coding sequences. Each RT-qPCR analysis was done from two to three independent biological samples in 3–4 technical replicates. Data were analyzed with the PikoReal software (version 2.2; Thermo Fisher) with manual threshold adjustment, and relative mRNA abundance was calculated in Microsoft Excel using the $\Delta\Delta C_T$ method. Statistical comparisons were calculated using the Student t-test.

## Acknowledgements

We are grateful to Roland Wedlich-Söldner, Claudine Kraft, Pierre Morsomme and Scott Emr for reagents. This work was supported by EMBO/Marie Curie (ALTF 642–2012; EMBOCOFUND2010, GA-2010–267146) and 'Tiroler Wissenschaftsfond' to OS, Austrian Science Fund (FWF-Y444-B12, P30263, P29583) and MCBO (W1101-B18) to DT and Agence Nationale pour la Recherche ('P-Nut', ANR-16-CE13-0002-01) to SL.

## Additional information

### Funding

| Funder | Grant reference number | Author |
|---|---|---|
| European Molecular Biology Organization | ALTF 642-2012 | Oliver Schmidt |
| European Molecular Biology Organization | EMBOCOFUND2010 | Oliver Schmidt |
| European Molecular Biology Organization | GA-2010-267146 | Oliver Schmidt |
| Tiroler Wissenschaftsfonds | 2015 | Oliver Schmidt |
| Austrian Science Fund | FWF-Y444-B12 | David Teis |
| Austrian Science Fund | P30263 | David Teis |
| Austrian Science Fund | P29583 | David Teis |
| Agence Nationale de la Recherche | ANR-16-CE13-0002-01 | Sebastien Leon |
| MCBO | W1101-B18 | David Teis |

The funders had no role in study design, data collection and interpretation, or the decision to submit the work for publication.

### Author contributions

Vasyl Ivashov, Resources, Formal analysis, Validation, Investigation, Visualization, Methodology, Writing - original draft; Johannes Zimmer, Resources, Data curation, Formal analysis, Validation, Investigation, Visualization, Methodology, Writing - original draft; Sinead Schwabl, Jennifer Kahlhofer, Resources, Validation, Investigation, Visualization, Methodology; Sabine Weys, Resources, Formal analysis, Investigation, Methodology; Ronald Gstir, Investigation, Methodology, Writing - review and editing, establishing and maintenance of the flow cytomerty screening platform; Thomas Jakschitz, Günther K Bonn, Investigation, Methodology, establishing and maintenance of the flow cytomerty screening platform; Leopold Kremser, Herbert Lindner, Formal analysis, Validation, Investigation; Lukas A Huber, Investigation, Methodology, Writing - original draft, establishing and maintenance of the flow cytomerty screening platform; Sebastien Leon, Resources, Formal analysis, Visualization,

Methodology, Writing - original draft, Writing - review and editing; Oliver Schmidt, Conceptualization, Resources, Data curation, Formal analysis, Supervision, Funding acquisition, Validation, Investigation, Visualization, Writing - original draft, Writing - review and editing; David Teis, Conceptualization, Resources, Data curation, Formal analysis, Supervision, Funding acquisition, Validation, Investigation, Visualization, Writing - original draft, Project administration, Writing - review and editing

## Author ORCIDs
Sebastien Leon (iD) http://orcid.org/0000-0002-2536-8595
Oliver Schmidt (iD) https://orcid.org/0000-0002-7921-4663
David Teis (iD) https://orcid.org/0000-0002-8181-0253

## Decision letter and Author response
Decision letter https://doi.org/10.7554/eLife.58246.sa1
Author response https://doi.org/10.7554/eLife.58246.sa2

## Additional files

### Supplementary files
• Supplementary file 1. Fluorescence microscopy-based screen for substrates of starvation-induced endocytosis.

• Supplementary file 2. Flow cytometry and fluorescence microscopy-based screen for genes specifically involved in starvation-induced endocytosis.

• Supplementary file 3. Gene ontology term analysis of 128 genes required for starvation-induced endocytosis of Mup1-pHluorin.

• Supplementary file 4. Primer for cloning and PCR-based genetic modifications.

• Transparent reporting form

### Data availability
All data generated or analysed during this study are included in the manuscript and supporting files.

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
