## [Decision Letter]

**Acceptance summary:**

The manuscript provides a deep and mechanistic understanding of a new response strategy to starvation in yeast. Specifically, it demonstrates that *Art2* has a role in selective endocytosis of several plasma membrane proteins, including amino acid transporters, during amino acid starvation. Importantly it uncovers the signals on the substrate and adaptor that are both necessary and sufficient for this process. We are excited to have it be published in *eLife*.

**Decision letter after peer review:**

Thank you for submitting your article "Complementary α-arrestin-ubiquitin ligase complexes control nutrient transporter endocytosis in response to amino acids" for consideration by *eLife*. Your article has been reviewed by three peer reviewers, one of whom is a member of our Board of Reviewing Editors, and the evaluation has been overseen by Suzanne Pfeffer as the Senior Editor. The reviewers have opted to remain anonymous.

The reviewers have discussed the reviews with one another and the Reviewing Editor has drafted this decision to help you prepare a revised submission.

Editorial summary and revision requests:

The manuscript by Ivashov et al. explores the molecular mechanisms leading to the selective endocytosis of amino acid transporters (AATs) in the baker's yeast *Saccharomyces cerevisiae* during starvation. It was known that AATs on the membrane need to be ubiquitinated by Rsp5 through adaptors called ARTs. While some insights into the substrate induced endocytosis (activated by enhanced substrate flux) have already emerged, it is unclear how endocytosis of AATs occurs during acute starvation. This manuscript is the first to characterize the signaling and regulatory pathway guiding endocytosis of an AAT during starvation and discovers a specific role for *Art2* as well as the signals on the substrate and adaptor that are both necessary and sufficient for this process. This paper is deep and mechanistic and uncovers a new regulatory cycle in a novel path of AAT starvation-dependent endocytosis and as such would be valuable to the cell biology community. Please answer the below requests prior to acceptance:

Experimental requests:

1) Is the transcriptional regulation enough to explain the differential requirement on ARTs for substrate induced vs. starvation induced endocytosis? I.e., if *Art1* is expressed under the *ART2* promoter in a *∆art2* during starvation – could it rescue the endocytosis of Mup1? This is not essential for publication, but would be a nice and easy experiment to do.

2) Figures 3B, 4A – how many times were they repeated? Please quantify the changes with SDs. This is true for all western blots whose claim is for a quantitative phenotype (increase or decrease in expression or in free GFP). Specifically, in Figure 4F quantification relative to the PGK control may show a similar increase in *GCN4^C^*…In addition, "WB analysis of total cell lysates indicated quantitative band shifts of *Art1*…". However, no quantification is provided. Please add quantification.

3) The authors use Mup1 (as one model substrate) and its downregulation as a criterion for adaptation to starvation. Does *Art2* deletion or the mutation in *Art2* (Figure 7B) cause any growth defect or problems with adaptation to starvation? This would be expected if downregulation via *Art2* is a long-term adaptation response.

4) The authors suggest that *Art1* and *Art2* act on different patches and provide convincing data along this line. Is it possible to generate chimeric constructs of *Art1* and *Art2* to swap their specificity? Since this may not work given that *Art2* has so many modification sites, this is not essential for resubmission.

Textual/Figure suggestions:

1) In the first Results section (subsection “Amino acid availability induces selective endocytosis of nutrient transporters”, first paragraph) when discussing the C' GFP tagging library of yeast please cite the manuscript that generated that library (PMID: 14562095) and not only in the Materials and methods.

2) The first part of the paper where C' GFP proteins were screened for expression is seems unrelated to the rest of the paper and not necessary for the future experiments. If the authors were interested specifically in AATs why did they not track all yeast AATs? It seems that some of them can be nicely visualized when tagged at their N' (such as Vba5, Thi72) or might need retagging and not to be taken from the library. Having only 7AATs to work with reduces the systematic nature of this study. Moreover, in the end they really only skim through the 4 well studied AATs (Mup1, Can, Lyp1 and Tat2) and really only focus on Mup1. Hence the Introduction is a bit misleading discussing not Mup1 but all AATs and making the first part of the paper quite unnecessary. Please reconsider writing the paper in a way that is more focused on Mup1 as a model and removing the above section.

3) Subsection “A genome-wide screen identifies regulators of starvation-induced endocytosis of AATs”, first paragraph: please change FACS to Flow Cytometry there and all places thereafter including Figure 2.

4) Subsection “A genome-wide screen identifies regulators of starvation-induced endocytosis of AATs”, second paragraph: typo.

5) Discussion, second paragraph: define UPS upon first use.

6) Figure 1A typo – should be methionine.

7) Figure 1B – "Ammonium Transporter" should be in color.

8) Figure 4B – what is n>x means – please write how many times exactly was it repeated.

9) Figure 6E has a very important piece of information, in that the authors show that the C-terminal segment of Mup1 can be transplanted to Hxt3. Please add the response of the un-modified Hxt3 in the same panel (and not in Figure 6—figure supplement 2A) to have the direct comparison.

10) The FACS-based screen of Mup1 traffic presented here is really quite nice, and provides a lot of information on proteins that regulate its endocytosis and vacuolar delivery. However, the finding that *Art2* regulates stress-dependent endocytosis of a variety of transporters at the PM is neither surprising nor novel: Nikko and Pelham, 2009, published a key paper that positioned *Art2* as a regulator of stress-dependent internalization of a wide variety of cargos. The stressor in was cycloheximide rather than starvation, suggesting a more general regulation of substrate-independent endocytosis. The Nikko and Pelham paper needs to be cited more prominently through this study. Its findings may not have the level of mechanistic detail presented here, but they do strongly parallel a number of key points.

11) The authors state that "low levels of methionine in the growth medium did not efficiently trigger its endocytosis." Supplementary Figure 1B complicates this interpretation, as there is clearly less Mup1 overall in 0.02 ug/ml Met compared to the no Met control. This decrease suggests that the protein indeed is being internalized, albeit at a slower rate. Flux from PM to vacuole could be slow enough that GFP fluorescence does not accumulate in the vacuole. Such a phenomenon was observed for Ptr2-GFP in Apel et al. (Mol Biol Cell 28: 2434-48). Please discuss this further in the manuscript.

12) Subsection “A genome-wide screen identifies regulators of starvation-induced endocytosis of AATs”, second paragraph: identify *pep4∆* as the sole "class 4" mutant, where Mup1-pHluorin accumulates in the vacuole, but remains fluorescent. This is unexpected, since Prosser et al., 2010, showed that Ste3-pHluorin is efficiently transported to the vacuole in *pep4∆prb1∆ prc1∆* cells, where fluorescence is still fully quenched. How might this discrepancy be explained?

13) The first paragraph of the subsection “Up-regulation of *Art2* by the GAAC pathway drives starvation-induced endocytosis of Mup1” suggests that the observed bandshift is due to Npr1-dependent phosphorylation. This is an over-interpretation, since no direct evidence of Npr1 involvement is shown. Moreover, since *Art1* is both phosphorylated and ubiquitinated, the band shift may not be due to phosphorylation at all. This should be more carefully interpreted, or more thoroughly investigated.

14) The first paragraph of the subsection “Up-regulation of *Art2* by the GAAC pathway drives starvation-induced endocytosis of Mup1” states that levels of *Art2*-HTF were "comparably" low under rich conditions. What is this compared to? The blot in Figure 4A shows an increase in *Art2* levels upon starvation, but levels of *Art1* appear to decrease. Please clarify.

15) Subsection “*Art2* directs Rsp5-dependent ubiquitination of C-terminal lysine residues in Mup1”, second paragraph: regarding the arrestin fold-localized PY motif, the authors state that it is required for starvation-induced endocytosis. This is an overstatement, since Figure 5E clearly shows Mup1 internalization when this PY motif is mutated, even though the amount of protein internalized is reduced. Please rephrase.

16) The first paragraph of the subsection “Up-regulation of *Art2* by the GAAC pathway drives starvation-induced endocytosis of Mup1” states that *Art1/2* regulate AAT endocytosis in a mutually exclusive manner, but this is not true for Tat2 based on Nikko and Pelham, 2008, where either is sufficient for Trp-induced internalization. Please discuss this.

17) Subsection “The C-terminal acidic sorting signal of Mup1 is sufficient for *Art2*-dependent starvation-induced endocytosis”, end of first paragraph: the callout for Supplementary Figure 6B should be for Figure 6—figure supplement 2D.

---

## [Author Response]

Experimental requests:1) Is the transcriptional regulation enough to explain the differential requirement on ARTs for substrate induced vs. starvation induced endocytosis? I.e., if Art1 is expressed under the ART2 promoter in a ∆art2 during starvation – could it rescue the endocytosis of Mup1? This is not essential for publication, but would be a nice and easy experiment to do.

Thank you for this suggestion. We have replaced the *ART1* promoter region with the *ART2* promoter region. The *ART2* promoter region also induced the expression of *Art1* during starvation (Figure 4—figure supplement 2A). Yet, the upregulation of *Art1* protein levels driven by the *ART2* promoter failed to restore starvation-induced endocytosis of Mup1-GFP in *art2∆* cells (Figure 4—figure supplement 2B), whereas the same construct was sufficient for substrate-induced endocytosis of Mup1-GFP in *art1∆* cells (Figure 4—figure supplement 2B).

These results are in agreement with our model of the switch from *Art1*-dependent to *Art2*-dependent amino acid transporter endocytosis during starvation. The results further show that the simple upregulation of *Art1* with the *ART2* promoter is not sufficient to initiate Mup1 endocytosis in response to starvation. Rather, the combination of *Art1* and *Art2* protein levels, the specific properties of *Art1* and *Art2* (e.g. their post-translational regulation, basic patches, etc…) as well as the exposure of the N- and C-terminal acidic patches on Mup1 define the substrate selectivity of endocytosis in response to nutrient availability.

2) Figures 3B, 4A – how many times were they repeated? Please quantify the changes with SDs. This is true for all western blots whose claim is for a quantitative phenotype (increase or decrease in expression or in free GFP). Specifically, in Figure 4F quantification relative to the PGK control may show a similar increase in GCN4^C^…In addition, "WB analysis of total cell lysates indicated quantitative band shifts of Art1…". However, no quantification is provided. Please add quantification.

We now provide quantification for the blots in Figure 3B (in Figure 3—figure supplement 1A), Figure 4 (in Figure 4—figure supplement 1B, C, F, G) and in Figure 6—figure supplement 2 (in Figure 6—figure supplement 2C, E). The results of the WB quantifications (including SD) confirm our original results. For the quantification, all blots have been repeated at least 3 times.

We agree with reviewers that the WB in Figure 4F was not ideally loaded to allow quantification. Therefore, we have repeated the experiment. The new result (Figure 4F), quantified in Figure 4—figure supplement 1G, confirms that constitutive Gcn4 translation (in *GCN4^C^*mutants) upregulates *Art2* already under rich growth conditions and that there is no further upregulation of *Art2* in *GCN4^C^* mutants during starvation.

3) The authors use Mup1 (as one model substrate) and its downregulation as a criterion for adaptation to starvation. Does Art2 deletion or the mutation in Art2 (Figure 7B) cause any growth defect or problems with adaptation to starvation? This would be expected if downregulation via Art2 is a long-term adaptation response.

Unicellular organisms must efficiently and robustly adjust cell growth and survival to nutrient availability. We have shown earlier that starvation-induced endocytosis and the selective lysosomal membrane protein degradation through the MVB pathway co-operates with autophagy to ensure cell survival during starvation (Müller et al., 2015).

In our current paper, we have built on this observation and have identified how selectivity of endocytosis is achieved to degrade a defined subset of plasma membrane (PM) proteins during starvation. The genome of *S. cerevisiae* encodes 14 different ART proteins that collectively function in the selective degradation of membrane proteins in response to different growth and stress conditions. We demonstrate that *Art2* is required to selectively target 5 different substrates (four amino acid transporters and Ina1) for starvation-induced endocytosis and degradation, out of at least 15 different membrane proteins. Other membrane proteins most likely will depend on other ARTs and will still be degraded in *art2∆* mutants, e.g. we demonstrate that starvation-induced endocytosis of Hxt3 requires Art4.

Hence, strong effects on starvation survival (similar to ESCRT mutants that block the lysosomal degradation of all membrane proteins – see Müller et al., 2015) could be expected by simultaneous deletion of multiple ART proteins or of the ubiquitin ligase Rsp5. However, *RSP5* is an essential gene, and mutants in which multiple *ART* genes are deleted display severe growth defects with loss of plasma membrane integrity due to proteotoxic stress (Nikko and Pelham, 2009; Zhao et al., 2013). The contribution of individual α-arrestins to survival might be relatively minor.

With this in mind, we have very carefully examined the growth and survival properties of *art2∆* mutants and compared them to isogenic WT cells, as well as to autophagy mutants (*atg8∆*) and *atg8∆ art2∆* double mutants. Indeed, the effects of *art2∆* mutants on cell growth are moderate. Under rich growth conditions and in logarithmic growth phase, *Art2* mutants grew slightly faster when compared to WT cells (Author response image 1), whereas the cell diameter was similar (Author response image 1).

**Author response image 1. respfig1:** Physiological effects of *ART2* deletion. (**A**) Growth of isogenic WT (*art2Δ* + *pART2)* and *art2Δ* cells in synthetic auxotrophy medium (YNB) at 26°C. Cultures were grown exponentially for 24 hours and then inoculated to OD_600nm_ = 0.1 at t=0h. Culture density was followed by OD_600nm_. Mean ± standard deviation from n = 3 independent experiments. (**B**) Mean cell diameter (± standard deviation) measured with CASY TTT cell counter (Omni Life Sciences, Bremen, Germany) of isogenic WT (*art2Δ* + *pART2*) and *art2Δ* cells grown exponentially for >24 hours and then starved for amino acids and nitrogen for the indicated time. (**C**) Isogenic WT (*art2Δ* + *pART2*), *art2Δ*, *atg8Δ* (*art2Δ*, *atg8Δ* + *pART2*) and *art2Δatg8Δ* cells grown exponentially for >24 hours and then starved for amino acids and nitrogen for the indicated time. Cells were stained with propidium iodide (PI, 3 μg/ml) for ten minutes, washed, and PI-positive cells were counted on an Attune TM NxT Cytometer (Life Technologies). 50000 cells were counted for each genotype and replicate. WT and *art2Δ*: mean ± standard deviation from n = 3 independent experiments. For the *atg8*Δ and *art2Δatg8Δ* cells two independent experiments are shown. (**D**) Isogenic WT (*art2Δ* + *pART2*), *art2Δ*, *atg8Δ* (*art2Δatg8Δ* + *pART2*) and *art2Δatg8Δ* cells were grown exponentially for >24 hours and then starved for amino acids and nitrogen (-N) for the indicated time. Survival was assessed by colony forming ability. Serial dilutions of the cultures were spotted on YPD rich medium. (**E**) Isogenic WT (*art2Δ* + *pART2*), *art2Δ*, *atg8Δ* (*art2Δatg8Δ* + *pART2*) and *art2Δatg8Δ* cells were grown exponentially for >24 hours and then starved for amino acids and nitrogen (-N) for 21 (upper panel) and 15 days (lower panel). Starved cells were re-inoculated into fresh auxotrophic selection medium (YNB) to OD_600nm_ = 0.1 at t = 0h. Culture density was followed by OD_600nm_.

When these cells were exposed to amino acid and nitrogen starvation for up to 21 days, *art2∆* mutants entered cell cycle arrest similar to WT cells (we scored the budding index after 24 hours and it was similar, data not shown) and displayed no significant differences in terms of cell diameter, survival (spot assays) or PM integrity / viability (measured by propidium iodide permeability) (Author response image 1). Yet, when *art2∆* cells were refed with fresh medium, after starving for 15 days or 21 days, they resumed growth slightly slower compared to WT cells (Author response image 1). As control we used autophagy mutants (*atg8∆*) that displayed defects in cell survival during starvation (Author response image 1, D) and failed to resume growth after starvation. We did not observe synthetic survival deficits of *art2∆* cells with autophagy mutants (*atg8∆*) during prolonged starvation (Author response image 1, D).

In conclusion, *ART2-*deficient cells appear to display mild deficits in adjusting cell growth both under rich conditions and while resuming growth after starvation. A precise mechanistic understanding of how *Art2* in concert with all other branches of the ART-Rsp5 ubiquitin ligase network contributes specifically to cell physiology under different growth conditions would require additional experiments (e.g. strain / nutrient competition experiments, metabolomics). We believe that these complex physiological experiments would go beyond the scope of the current study, which focuses on the mechanism for selective starvation-induced endocytosis. Therefore, at the moment we think it would be premature to include these new experiments in the paper.

Consistent with the results presented here, another recent study (Wloch-Salamon DM et al., G3 (Genes, Genomes, Genetics), 2017, PMID: 28450371) found that over many repeated growth-starvation cycles (equivalent to 300 generations) mutations in *Art2* (Ecm21) and Rsp5 affected survival in limiting nutrients. These findings also point to a physiological function of *Art2* during starvation. However, it is not clear how these physiological defects are linked to the role of *Art2* in starvation induced endocytosis and we now discuss this and cite the study by Wloch-Salamon et al., 2017.

4) The authors suggest that Art1 and Art2 act on different patches and provide convincing data along this line. Is it possible to generate chimeric constructs of Art1 and Art2 to swap their specificity? Since this may not work given that Art2 has so many modification sites, this is not essential for resubmission.

This would be indeed a great experiment. At the moment, however, this experiment has little chance of success, due to the above mentioned massive post-translational modifications of both proteins and the seemingly inverse activation / inactivation of *Art1* and *Art2*.

Moreover, recent work from the Emr lab suggests that the activity of the α-arrestins is regulated by disordered loops in their arrestin domains (Baile et al., 2019). The disordered loop insertion in *Art2* appears to be more extensive compared to *Art1* (see schematic in Author response image 2). In *Art2*, a clear signature for an arrestin-N domain cannot be detected and the arrestin-C domain is extended to more than twice the length of the arrestin-C pfam model (Baile et al., 2019). In addition, these disordered loops are in the different positions in the arrestin N- and C- domains of *Art1* and *Art2*, respectively, and therefore render precise domain swap experiments almost impossible without additional structural information.

**Author response image 2. respfig2:** Schematic depicting the primary structure of *Art1* and *Art2* with predicted arrestin-N (orange) and arrestin-C (yellow) domains and the position of interspersed, disordered loops (dark grey), basic patches (blue) and PY motifs (red) drawn according to Baile et al., 2019).

Textual/Figure suggestions:1) In the first Results section (subsection “Amino acid availability induces selective endocytosis of nutrient transporters”, first paragraph) when discussing the C' GFP tagging library of yeast please cite the manuscript that generated that library (PMID: 14562095) and not only in the Materials and methods.

Thank you for pointing this out. We have now included the corresponding reference in the Results section of the manuscript.

2) The first part of the paper where C' GFP proteins were screened for expression is seems unrelated to the rest of the paper and not necessary for the future experiments. If the authors were interested specifically in AATs why did they not track all yeast AATs? It seems that some of them can be nicely visualized when tagged at their N' (such as Vba5, Thi72) or might need retagging and not to be taken from the library. Having only 7AATs to work with reduces the systematic nature of this study. Moreover, in the end they really only skim through the 4 well studied AATs (Mup1, Can, Lyp1 and Tat2) and really only focus on Mup1. Hence the Introduction is a bit misleading discussing not Mup1 but all AATs and making the first part of the paper quite unnecessary. Please reconsider writing the paper in a way that is more focused on Mup1 as a model and removing the above section.

We agree with the reviewers that the analysis presented in Figure 1 is neither systematic nor comprehensive and we have toned down the results and adapted the text of Figure 1 accordingly.

Yet, we think that the results of Figure 1 are important (i) because they demonstrate in an unprecedented way the scale and the selectivity of starvation-induced endocytosis towards certain cargoes and (ii) they are important to delineate the specificity of *Art2* for a subset of cargoes including Ina1 (new result Figure 3—figure supplement 1C), and four different amino acid transporters (AAT), but not for all membrane proteins that undergo starvation-induced endocytosis (e.g. the glucose transporters) (Figure 3—figure supplement 1D). Therefore, we prefer to keep the data of Figure 1 in the paper.

We have now also identified a further membrane protein (Ina1), which undergoes *Art2*-dependent starvation-induced endocytosis (Supplementary file 1; Figure 3—figure supplement 1C). Ina1 is a protein of unknown function with three predicted transmembrane domains and a cytoplasmic C-terminus and not related to AATs, suggesting that the action of *Art2* during starvation is not limited to AATs. The Ina1 sequence also features a putative acidic patch (included in Figure 6D) in its C-terminus, and its starvation-induced endocytosis is impaired in the basic patch mutant *Art2* K664D,R665D,R666D,K667D (Figure 7—figure supplement 1B).

In Figure 1, we have screened a library containing 149 putative PM proteins that were C-terminally GFP-tagged at their native chromosomal locus. According to a recent and comprehensive review by Bert Poolman and colleagues (Bianchi et al., 2019), the *S. cerevisiae* S288C genome encodes 21 AATs that have been reported to localize to the plasma membrane (we have corrected the number accordingly in the Introduction). Our library contained 16 of these 21 AATs, and thus covers 76% of all AATs that are reported to localize to the PM and lacks five (Agp3, Alp1, Hip1, Put4, and Vba5) (see modified Supplementary file 1).

Of the 16 AATs represented in our library, we detect under rich growth conditions (YNB medium, genetic background BY4741) eight different AATs (Mup1, Can1, Tat2, Lyp1, Dip5, Gnp1, Bap2 and Tat1; see modified Supplementary file 1) at the PM. In the original version of the manuscript we have miscounted seven AATs. We apologize for this mistake and have corrected the numbers accordingly. A ninth AAT, Gap1-GFP, was upregulated during starvation. We did not detect GFP signals for Agp1, Agp2, Yct1, Bap3, Mup3 or Mmp1 in our rich and starvation media. The signal for Sam3-GFP was detected but mislocalized to the ER.

Perhaps it should not come as a surprise that we do not detect all AATs under rich growth conditions. This is exemplified by Gap1 which is only expressed during starvation, or by Put4, which is repressed by ammonium as nitrogen source (Jauniaux, et al.,1987, Eur. J. Biochem.). Earlier studies addressed systematically the localization of N- and C-terminally GFP-tagged proteins expressed from their native promoters to generate the localization and quantitation atlas of the yeast proteome (in the Loqate database) (Breker et al., 2013, J. Cell Biol.; Breker et al., 2013, NAR). Even these studies could not report the localization of all putative AATs at the PM. Agp1, Agp2, Agp3, Alp1, Bap3, Mmp1, Put4, Sam3, Mup3, Ydt1, Vba5 (i.e. 11 AATs out of 21) were either not detected or were localized to punctate structures or the nuclear ER.

To the best of our knowledge N- or C-terminally tagged versions of Vba5 were over-expressed to detect PM localization (Shimazu et al., 2012, Biosci. Biotechnol. Biochem.). The thiamine (vitamin B1) transporters Thi7, Nrt1 (Thi71) and Thi72 have been proposed to be *Art2*-dependent cargoes during substrate excess (Savocco et al., 2019). In this publication, Nrt1 and Thi72 were only detected at the PM when they were expressed under the *THI7* promoter, and remained functional for thiamine transport upon N- or C-terminal tagging.

Therefore, we tested the role of *Art2* in Thi7-eGFP, Nrt1-eGFP and Thi72-eGFP endocytosis during starvation. In agreement with our initial screen, Thi7-eGFP was not detected at the PM in the majority of cells, presumably because its expression is repressed by thiamine included in our growth medium (Hohmann and Meacock, 1998, BBA).

Nrt1-eGFP and Thi72-eGFP (both expressed from the *THI7* promoter) localized to the plasma membrane (and vacuole) in growing cells. Both proteins underwent efficient, *Art2*-dependent starvation-induced endocytosis, suggesting that the extent of *Art2*-dependent membrane protein turnover during starvation may also vary with the growth conditions. Yet, since these proteins were not expressed from their native promoters, we prefer not to include these data into the paper and only present them here for the reviewers’ convenience (Author response image 3).

**Author response image 3. respfig3:** Life cell fluorescence microscopy of the indicated strains chromosomally expressing. Thi7-eGFP, or expressing Thi-eGFP or Nrt1-eGFP from centromeric plasmids under control of the *THI7* promoter (Savocco et al., 2019) during exponential growth and after 6 hours of amino acid and nitrogen starvation (-N).

3) Subsection “A genome-wide screen identifies regulators of starvation-induced endocytosis of AATs”, first paragraph: please change FACS to Flow Cytometry there and all places thereafter including Figure 2.

Done – thank you.

4) Subsection “A genome-wide screen identifies regulators of starvation-induced endocytosis of AATs”, second paragraph: typo.

Unfortunately, we could not spot a typo in the subsection “A genome-wide screen identifies regulators of starvation-induced endocytosis of AATs”. In general, we have tried our best to avoid and correct typos.

5) Discussion, second paragraph: define UPS upon first use.

Done – thank you.

6) Figure 1A typo – should be methionine.

Done – thank you.

7) Figure 1B – "Ammonium Transporter" should be in color.

We now colored it purple.

8) Figure 4B – what is n>x means – please write how many times exactly was it repeated.

Thank you for pointing this out. In Figure 4B we have now indicated the exact numbers of repetitions.

9) Figure 6E has a very important piece of information, in that the authors show that the C-terminal segment of Mup1 can be transplanted to Hxt3. Please add the response of the un-modified Hxt3 in the same panel (and not in as Figure 6—figure supplement 2A) to have the direct comparison.

Thank you for this suggestion. We have modified Figure 6 accordingly for direct comparison.

10) The FACS-based screen of Mup1 traffic presented here is really quite nice, and provides a lot of information on proteins that regulate its endocytosis and vacuolar delivery. However, the finding that Art2 regulates stress-dependent endocytosis of a variety of transporters at the PM is neither surprising nor novel: Nikko and Pelham, 2009, published a key paper that positioned Art2 as a regulator of stress-dependent internalization of a wide variety of cargos. The stressor in was cycloheximide rather than starvation, suggesting a more general regulation of substrate-independent endocytosis. The Nikko and Pelham paper needs to be cited more prominently through this study. Its findings may not have the level of mechanistic detail presented here, but they do strongly parallel a number of key points.

Thank you, for finding the results of our screen interesting. We hope that it will represent a new and valuable resource to study how cells changes during nutrient limitation.

We also agree that the Nikko and Pelham, 2009 paper, was important and we are now citing the paper prominently on six different occasions throughout the paper. Yet, we are equally convinced that our findings on the non-redundant role and the regulation of *Art2* for starvation-induced endocytosis of Mup1, Can1, Lyp1 and Tat2 could not have been predicted based on the Nikko and Pelham paper. This is based on the following arguments:

1) It is difficult (if not impossible) to compare on a mechanistic and physiological level cellular stress that is induced by cycloheximide (CHX) treatment to amino acid starvation. CHX is a translation inhibitor and therefore indirectly boosts the levels of intracellular amino acids (Beugnet, A., Biochem. J. 2003, Urban J., Mol. Cell 2007). This results in TORC1 hyperactivation (Binda M., et al. Mol. Cell 2009) and in acute growth arrest. In contrast amino acid starvation results in a depletion of free intracellular amino acid pools and in TORC1 inactivation. However, cells starving for amino acid still complete their cell cycle and slowly and ‘orderly’ enter in a G1/G0 cell cycle arrest to survive for several weeks.

2) In the Nikko paper *Art2* typically has a redundant function with other α-arrestins. Expression of *Art1*, *Art2*, *Art8* and Bsd2 in the 9-arrestin deletion strain partially restored CHX-induced endocytosis of the uracil transporter Fur4 (Figure 4B; Nikko and Pelham, 2009). Expression of *Art2*, *Art8* and Bsd2 partially restored CHX-induced endocytosis of Tat2 in the 9-arrestin deletion strain (Figure 6C Nikko and Pelham, 2009).

3) The Nikko paper did not address the role of *Art2* for Mup1, Lyp1 and Can1 endocytosis, which is the main focus of our work. In our hands, endocytosis of these AATs in response to CHX treatment was not (Mup1, Can1) or only partially (Lyp1) dependent on *Art2*. Endocytosis of Mup1 in response to CHX was limited (after 6 h + CHX, a large pool of Mup1-GFP still localized to the PM with a fraction detected inside the vacuole) and this was dependent on the presence of *Art1* (see Author response image 4). Based on these results, it seems that *Art2* has a minor (and potentially redundant) role for the CHX-induced endocytosis of Mup1, Can1 and Lyp1. This is in stark contrast to our results which show a rather stringent *Art1* / *Art2* – switch for AAT endocytosis in response to amino acid excess (*Art1*) or amino acid starvation (*Art2*).

We prefer not to include these results using CHX as a stressor, since they do not add significant new information on the regulation of nutrient transporter endocytosis in response to nutrient availability.

**Author response image 4. respfig4:** Life cell fluorescence microscopy of WT, *art2Δ* or *art1Δ* cells chromosomally expressing Mup1-GFP (A), Can1-GFP (B) or Lyp1-GFP (C) during exponential growth and 6 hour after treatment with 50 μg/ml cycloheximide (+CHX) or vehicle (water; -CHX).

11) The authors state that "low levels of methionine in the growth medium did not efficiently trigger its endocytosis." Supplementary Figure 1B complicates this interpretation, as there is clearly less Mup1 overall in 0.02 ug/ml Met compared to the no Met control. This decrease suggests that the protein indeed is being internalized, albeit at a slower rate. Flux from PM to vacuole could be slow enough that GFP fluorescence does not accumulate in the vacuole. Such a phenomenon was observed for Ptr2-GFP in Apel et al. (Mol Biol Cell 28: 2434-48). Please discuss this further in the manuscript.

Thank you for pointing this out. In our original microscopy picture the overall Mup1-GFP fluorescence appeared rather low, with the Mup1-GFP signal mainly detected at the PM. This may have caused the impression that Mup1-GFP may be slowly turned over.

Therefore, we have carefully re-analyzed the data and performed new experiments. We found that in medium containing 0.02 µg/ml methionine Mup1 endocytosis was not yet triggered, at least on a detectable level. The protein levels of Mup1-GFP were analyzed by Western blot analysis from cells grown in medium without methionine or with different methionine concentrations. Without methionine or with low levels of methionine (0.02 µg/ml) the protein levels of full length Mup1-GFP were similar (new Figure 1—figure supplement 2B), whereas only clipped GFP was detected in response 2 µg/ml methionine or starvation (new Figure 1—figure supplement 2B). Our original microscopy picture did not very well represent the data. The overall Mup1-GFP appeared to be low, but localized almost exclusively to the PM. Hence it is likely that its fluorescence intensity of was not well adjusted. We apologize for the mistake and we have exchanged the image in Supplementary Figure 1B (now called Figure 1—figure supplement 2A) for a representative image that also reflects the results of the new western-blot analysis (new Figure 1—figure supplement 2B).

12) Subsection “A genome-wide screen identifies regulators of starvation-induced endocytosis of AATs”, second paragraph: identify pep4∆ as the sole "class 4" mutant, where Mup1-pHluorin accumulates in the vacuole, but remains fluorescent. This is unexpected, since Prosser et al., 2010, showed that Ste3-pHluorin is efficiently transported to the vacuole in pep4∆ prb1∆ prc1∆ cells, where fluorescence is still fully quenched. How might this discrepancy be explained?

The fluorescence of Mup1-pHlorin is partially quenched inside the vacuoles of *pep4∆* single or *pep4∆ prb1∆ prc1∆* triple mutants in response to starvation (see Author response image 5, this is not a major focus of our work).

One simple explanation for the difference of Mup1 to Ste3 might be the difference in their abundance. Mup1 is among the most highly expressed PM proteins in medium lacking methionine (Busto et al., 2018, PNAS, Figure 1), and it is probably vastly more abundant compared to Ste3. Hence, ‘unquenched’ Mup1-pHluorin fluorescence could be easily detected. Moreover, during starvation vacuolar enzymes are additionally engaged in the degradation of autophagic cargoes. Therefore, the breakdown of intraluminal vesicles that are delivered via the MVB pathway may become more challenging in *pep4∆* mutants during starvation. Similarly, we also observed earlier that the autophagy protein pHluorin-*Atg8* was not efficiently quenched inside the vacuole in the majority of *pep4∆ prb1∆ prc1∆* cells during nitrogen starvation (Müller et al., 2015, Figure 3E, F).

**Author response image 5. respfig5:** Impaired pHluorin quenching in *pep4Δ* cells. (**A**) Flow cytometry analysis of BY4742 WT (green histogram), *art2Δ* (yellow) and *pep4Δ* cells (red) after 20h of amino acid and nitrogen starvation. (**B**) Live cell fluorescence microscopy of BY4742 WT and *pep4Δ* cells during exponential growth, after 24h of amino acid and nitrogen starvation (-N), and after 90 minutes of methionine treatment (20 μg/ml; +Met). Fluorescence images of WT and *pep4Δ* cells were equally adjusted to allow for comparison of GFP fluorescence intensity. (**C**) Live cell fluorescence microscopy of SEY6210 WT and *pep4Δprb1Δprc1Δ* cells analyzed as in B).

13) The first paragraph of the subsection “Up-regulation of Art2 by the GAAC pathway drives starvation-induced endocytosis of Mup1” suggests that the observed bandshift is due to Npr1-dependent phosphorylation. This is an over-interpretation, since no direct evidence of Npr1 involvement is shown. Moreover, since Art1 is both phosphorylated and ubiquitinated, the band shift may not be due to phosphorylation at all. This should be more carefully interpreted, or more thoroughly investigated.

We thank the reviewers for pointing this out. This comment actually prevented us from making a mistake. For quantification we have now repeated this experiment three times. When repeating the experiment, we could no longer detect the band-shifts for *Art1* and therefore we have replaced Figure 4A with the new experiments including quantification in Figure 4—figure supplement 1B. We now rephrased this section, as recommended, and omitted all statements regarding band-shift of *Art1* from the Results section.

‘WB analysis of total cell lysates indicated that 6xHis-TEV-3xFlag-tagged *Art1* protein levels were unchanged in response to starvation (Figure 4A, lanes 2,3), whereas the protein levels of the functional 170kD protein *Art2*-HTF (Figure 4—figure supplement 1A) were up-regulated in response to starvation (Figure 4A, lanes 5,6; Figure 4—figure supplement 1B).’

14) The first paragraph of the subsection “Up-regulation of Art2 by the GAAC pathway drives starvation-induced endocytosis of Mup1” states that levels of Art2-HTF were "comparably" low under rich conditions. What is this compared to? The blot in Figure 4A shows an increase in Art2 levels upon starvation, but levels of Art1 appear to decrease. Please clarify.

We have re-phrased the corresponding sentence as follows:

‘WB analysis of total cell lysates indicated that 6xHis-TEV-3xFlag-tagged *Art1* protein levels were unchanged in response to starvation (Figure 4A, lanes 2,3), whereas the protein levels of the functional 170kD protein *Art2*-HTF (Figure 4—figure supplement 1A) were up-regulated in response to starvation (Figure 4A, lanes 5,6; Figure 4—figure supplement 1B).’

15) Subsection “Art2 directs Rsp5-dependent ubiquitination of C-terminal lysine residues in Mup1”, second paragraph: regarding the arrestin fold-localized PY motif, the authors state that it is required for starvation-induced endocytosis. This is an overstatement, since Figure 5E clearly shows Mup1 internalization when this PY motif is mutated, even though the amount of protein internalized is reduced. Please rephrase.

We have rephrased this section as follows:

‘*Art2* has four putative PY motifs (Figure 5C). Mutations in the PY motif that resides within the predicted arrestin fold of *Art2* (P748,P749,Y750) reduced starvation induced endocytosis of Mup1-GFP (Figure 5E). The *Art2*^P748A,P749A,Y750A^ mutant was expressed at similar levels than the WT protein (Figure 5—figure supplement 1B). We suggest that interaction between WW3 in Rsp5 and the PY motif (748-750) of *Art2* was required for the efficient starvation-induced endocytosis' of Mup1.’

16) The first paragraph of the subsection “Up-regulation of Art2 by the GAAC pathway drives starvation-induced endocytosis of Mup1” states that Art1/2 regulate AAT endocytosis in a mutually exclusive manner, but this is not true for Tat2 based on Nikko and Pelham, 2008, where either is sufficient for Trp-induced internalization. Please discuss this.

We mention this in the text:

‘Substrate-induced endocytosis of Mup1, Can1 and Lyp1 required *Art1*, whereas in case of Tat2 substrate-induced endocytosis was less stringent and could be mediated either by *Art1* or *Art2* (Nikko and Pelham, 2009).’

We additionally specify:

‘Our experiments so far demonstrated that *Art1* and *Art2* mediate the endocytic down-regulation of Mup1, Can1 and Lyp1 in response to changes in amino acid availability in a mutually exclusive manner.’

In addition, we indicate that Tat2 differs from Mup1, Can1 and Lyp1:

‘In addition, the *Art2* basic patch mutant also impaired, at least partially, starvation-induced endocytosis of Can1, Lyp1 and Ina1, while the endocytosis of Tat2 was independent of the basic patch (Figure 7B, Figure 7—figure supplement 1B, C).

Overall, it seemed that *Art2* employed a positively charged region in its arrestin fold to recognize C-terminal acidic patches in at least three different AATs (Mup1, Can1, Lyp1) and thus mediate their endocytosis in starvation conditions.’

17) Subsection “The C-terminal acidic sorting signal of Mup1 is sufficient for Art2-dependent starvation-induced endocytosis”, end of first paragraph: the callout for Supplementary Figure 6B should be for Figure 6—Figure supplement 2D .

Thank you. We have corrected this mistake.

Additional note: We would like to point the reviewers’ attention to the fact that we have replaced two images of exponentially growing cells expressing Mup1-GFP (Figure 4E, left panel; Figure 1—figure supplement 2A, left panel). We noted that we had mixed them up accidentally with similar images during the figure composition. Both panels were now exchanged for the correct pictures that correspond to these experiments. Of course, these changes do not affect any of our conclusions. We apologize for this mistake.